# Unveiling the Coexistence of Generalization and Memorization: A Case Study on Arithmetic Tasks with Label Noise

## Abstract

Highly over-parameterized models can simultaneously memorize noisy labels and generalize well, yet how these behaviors coexist remains poorly understood. In this work, we investigate the underlying mechanisms of this coexistence using modular arithmetic tasks under heavy label noise. Through extensive experiments on two-layer neural networks, we find that larger models tend to generalize better under appropriate optimization and model configurations, while noisy labels are memorized faster than clean data. Over-parameterized models internally form a generalization structure, but its expression in the output is suppressed by the need to fit noisy labels. Remarkably, even with 80% label noise, near-perfect test accuracy can be achieved by extracting this internal structure using frequency-based methods. We further propose a task-agnostic method to partition networks into generalization and memorization components. Although this subnetwork improves generalization, it is limited compared with frequency-based extraction, indicating that the generalization structure is distributed across neurons and motivating the development of new tools to retrieve generalizable knowledge from over-parameterized networks.

## 1. Introduction

The remarkable success of Large Language Models (LLMs) is closely associated with the empirical observation that increasing model capacity often leads to improved generalization. (Brown et al., 2020; Kaplan et al., 2020; Henighan et al., 2020; Hoffmann et al., 2022). In various structured tasks, larger models often exhibit emergent capabilities, demonstrating a robust ability to capture underlying rules from data (Wei et al., 2022a;b; Srivastava et al., 2023). However, in practical applications, training corpora are frequently contaminated with noise, such as mislabeled samples or factual inconsistencies. While the scaling paradigm suggests that "bigger is better" in clean environments (Liu et al., 2025), it remains an open question how over-parameterized models capture underlying rules from corrupted data, and where the boundaries of this inherent robustness lie.

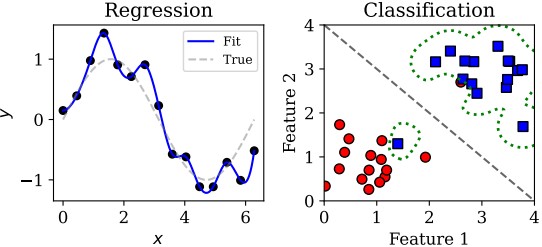

*Figure 1.* Illustration of label noise in regression and classification tasks. In regression, an over-parameterized model smoothly interpolates training samples. In classification, generalization can arise either from not fitting noisy labels (gray curve) or from fitting them while maintaining a smooth decision boundary (green curve).

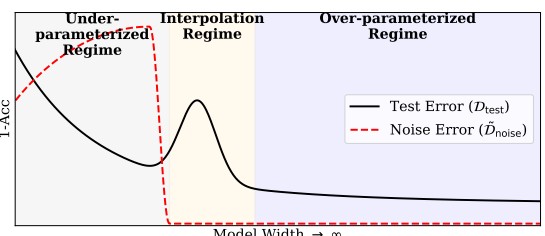

*Figure 2.* Illustration of model-wise double descent. Beyond the interpolation threshold, the over-parameterized regime corresponds to the second descent, where increasing model size leads to improved generalization, even after fitting all training samples, including label noise.

A critical yet poorly understood aspect of this problem is the coexistence of generalization and memorization in classification tasks under label noise. Unlike regression settings, where noisy labels often still carry ground-truth information (Montanari & Urbani, 2025), classification noise represents a categorical error that provides no informative signal for the

[1]Anonymous Institution, Anonymous City, Anonymous Region, Anonymous Country. Correspondence to: Anonymous Author <anon.email@domain.com>.

Preliminary work. Under review by the International Conference on Machine Learning (ICML). Do not distribute.

decision boundary (see Figure 1). While classical statistical learning suggests that strong regularization can mitigate overfitting by pushing models into the under-parameterized regime (Liu et al., 2022a), modern deep learning reveals a different story through the lens of double descent (Figure 2) (Belkin et al., 2019; Nakkiran et al., 2021). Empirical observations show that even with a significant proportion of label noise (e.g., 20%), increasing model size beyond the interpolation threshold allows the network to maintain high generalization while simultaneously memorizing the noise (Somepalli et al., 2022; Gamba et al., 2022; Xu et al., 2023). However, the mechanism of this phenomenon is not yet fully understood, especially in regimes of heavy label noise, where the pressure to memorize may conflict more severely with the drive to generalize. It remains unclear whether the "bigger is better" paradigm holds under extreme corruption and how the model internally resolves this competition. This gap leads us to study

Q1. Under what conditions does scaling model size consistently improve generalization under label noise?

Q2. Can over-parameterized models internally recover structural rules even under extreme label noise?

Q3. Can generalization and memorization be effectively disentangled at the neuronal level?

To address these questions, we study two-layer neural networks (NN), the cleanest non-linear NNs, on modular arithmetic tasks (addition, subtraction, and multiplication modulo $P$). This setup is particularly suitable as it exhibits the well-known *grokking* phenomenon, whose underlying mechanisms have been extensively studied in noise-free settings (Power et al., 2022; Nanda et al., 2023). More importantly, modular tasks allow for rigorous theoretical treatment, with proven analytical solutions for models using quadratic activations (Morwani et al., 2024; Tian, 2025). By introducing label noise, we create a challenging environment to investigate the competition between noise memorization and rule discovery. We conduct extensive controlled experiments across diverse configurations, including varying optimizers, weight decays, and model structures. We identify consistent generalization patterns across networks with different activations. Further analysis reveals that these networks learn the generalization internally even under extreme noise. Moreover, we propose a neuron-based network partitioning method to evaluate the separability of these learned components. Our main findings are summarized as follows:

1. We show that the "bigger is better" trend holds in the over-parameterized regime of double descent, provided that optimization, regularization, and model structure are properly configured.

2. Through a dynamical analysis, we reveal a counter-intuitive learning order where noisy labels are memo-

rized faster than clean data, which holds consistently across diverse optimization configurations.

3. We find that models can internally learn structural rules even under 80%∼90% noise, but their expression is suppressed by the need to memorize noisy labels.

4. We demonstrate that while network partitioning improves generalization performance, it cannot fully isolate generalization from memorization due to their complicated entanglement.

These results highlight the limitations of current approaches in fully extracting the knowledge learned by over-parameterized models, suggesting the need for new tools and learning paradigms. The rest of the paper will elaborate on the above takeaways. In Section 2, we introduce the problem setup. In Section 3, we discuss the conditions for the occurrence of the over-parameterized regime. In Section 4, we present the internal generalization representation. In Section 5, we propose and validate the network partition methods for separating generalization and memorization. We defer the related work to Appendix A.

## 2. Problem Setup

We study a modular arithmetic task $a \circ b = c$ following Power et al. (2022), where $a, b, c \in \{0, \ldots, P-1\}$ and $\circ$ denotes a modular operation such as modular addition, subtraction, or multiplication. We focus primarily on modular addition and use other operations to test generality. The task is formulated as a $P$-class classification problem.

We train a two-layer neural network parameterized by $\theta$:

$$\boldsymbol{f}_\theta(a, b) = \boldsymbol{W}\,\phi(\boldsymbol{U}\boldsymbol{e}_a + \boldsymbol{V}\boldsymbol{e}_b) + \boldsymbol{\mu}, \tag{1}$$

where $\theta = \{\boldsymbol{W}, \boldsymbol{U}, \boldsymbol{V}, \boldsymbol{\mu}\}$. Here, $\boldsymbol{e}_a, \boldsymbol{e}_b \in \mathbb{R}^P$ are one-hot encodings of $a$ and $b$; $\boldsymbol{U}, \boldsymbol{V} \in \mathbb{R}^{M \times P}$ are first-layer weights; $\boldsymbol{W} \in \mathbb{R}^{P \times M}$ is the second-layer weight matrix; $\boldsymbol{\mu} \in \mathbb{R}^P$ is the bias; and $M$ denotes the number of hidden neurons. The output $\boldsymbol{f}_\theta(a, b) \in \mathbb{R}^P$ is trained to predict the one-hot label $\boldsymbol{e}_{a \circ b}$.

Unless stated otherwise, we fix $P = 113$ following Nanda et al. (2023) and vary the model width as $M = 2^k + 1$ for $k \in \{4, \ldots, 12\}$, since ReLU models empirically generalize better at odd widths in these tasks (Pearce et al., 2023).

**Dataset.** The total dataset is

$$\mathcal{D}_{\text{total}} = \{(a, b, c) : a, b \in \{0, \ldots, P-1\},\ c = a \circ b\}.$$

We randomly split $\mathcal{D}_{\text{total}}$ into training, validation, and test sets with ratios 50%, 20%, and 30%, respectively. In Appendix C.1, we also vary the training ratio to 40% and 60% while fixing the test ratio.

To introduce label noise, we randomly select a subset $\mathcal{D}_{\text{noise}} \subset \mathcal{D}_{\text{train}}$ with noise ratio $\alpha$, i.e., $\frac{|\mathcal{D}_{\text{noise}}|}{|\mathcal{D}_{\text{train}}|} = \alpha$. For each $(a, b, c) \in \mathcal{D}_{\text{noise}}$, we define the observed label $\tilde{c}$ as

$$\tilde{c} = \text{Uniform}\big(\{0, \cdots, P - 1\} \setminus \{c\}\big), (a, b, c) \in \mathcal{D}_{\text{noise}}.$$

The resulting noisy training subset is given by

$$\tilde{\mathcal{D}}_{\text{noise}} := \{(a, b, \tilde{c}) : (a, b, c) \in \mathcal{D}_{\text{noise}}\}.$$

The noisy training set is $\tilde{\mathcal{D}}_{\text{train}} = \mathcal{D}_{\text{clean}} \cup \tilde{\mathcal{D}}_{\text{noise}}$. All evaluations are performed on clean validation and test sets.

**Evaluation.** To measure the performance of a model $\boldsymbol{f}$, we define its accuracy on a dataset $\mathcal{D}$ as

$$\text{Acc}(\boldsymbol{f}, \mathcal{D}) = \frac{1}{|\mathcal{D}|} \sum_{(a,b,c) \in \mathcal{D}} \mathbb{I}(\hat{c} = c),$$

where $\hat{c} = \arg\max_{c'} [\boldsymbol{f}(a, b)]_{c'}$, and $\mathbb{I}(\cdot)$ is the indicator function.

**Optimizers.** We train models with cross-entropy loss using Adam, AdamW (Loshchilov & Hutter, 2017), and Muon (Jordan et al.), with weight decay controlling regularization strength. We also experiment with SGD, but it is inefficient and requires careful hyperparameter tuning to achieve generalization (see Figure 21).

Unless specified otherwise, all results are reported after 200,000 epochs of full-batch training. We apply a linear learning-rate warmup over the first 10 steps. The default setting uses ReLU activations with AdamW (learning rate $10^{-3}$, weight decay 0.1, random seed 42) on the modular addition task.

## 3. When Bigger Is Better

In this section, we investigate how the presence of label noise interacts with model capacity in modular arithmetic tasks. We approach this question from two complementary perspectives. First, we examine end-of-training performance, asking under which conditions—such as optimization and regularization choices—increasing model size consistently improves generalization despite noise. Second, we analyze the training dynamics to study whether the model learns clean labels and noisy labels at different rates.

### 3.1. End-of-Training Performance

We robustly observe the double descent of the test *error* across different activation functions (Figure 4), noise-to-training ratios (Figure 16 and 17), optimizers (AdamW: Figure 3, Adam: Figure 18, Muon: Figure 19), and arithmetic tasks (Figure 20). In the over-parameterized regime, where

models fully memorize both the clean and noisy training data, continually enlarging the model creates non-decreasing test accuracy, but the threshold of model size for entering the over-parameterized regime depends on the regularization. We summarize the observations in Result 3.1.

**Result 3.1.** *Double descent is consistently observed across varying setups:*

1. *Within the over-parameterized regime, bigger models generalize better.*

2. *Strengthening regularization will shift models from the over-parameterized regime toward the under-parameterized regime. (Figure 3)*

3. *The model sizes at which test accuracy saturates vary across different activation functions. (Figure 4)*

Regularization does play a role in shaping the double-descent curve by controlling the effective model complexity (EMC) (Nakkiran et al., 2021). Intuitively, stronger regularization restricts the set of functions that can be realized during training, effectively reduce the model's capacity. When the EMC is limited, learning signals for generalization and memorization compete more strongly. As a result, increasing regularization can be beneficial when memorization of label noise is undesirable.

Models with quadratic or GeLU activation functions exhibit clear ceilings in test accuracy as model size increases within the explored model size range (width $< 10,000$, Figure 4). We include quadratic activations here because their analytical solutions are available (see Section 4). One possible explanation for these ceilings is that the symmetry of these activations constrains how noise can be absorbed while preserving generalization. In contrast, ReLU activations store noise in a non-symmetric manner, while the representation of generalization remains symmetric (see Section 4), allowing model capacity to be used more flexibly and leading to improved generalization.

We also observe that models are more confident (i.e., exhibit lower test cross-entropy loss) in the over-parameterized regime than in the under-parameterized regime. Larger models are generally more confident (Figure 3, bottom row). Cross-entropy loss reflects the confidence calibration of model predictions. In the under-parameterized regime, models often predict the true label even for corrupted samples. To align the predicted probability with the portion of correctness, the model assigns lower probabilities, resulting in higher uncertainty and higher test cross-entropy. In contrast, in the over-parameterized regime, larger models can separate data with larger margins while still satisfying the same $\ell_2$ regularization constraint. Larger margins lead to more confident predictions and, consequently, lower test cross-entropy loss.

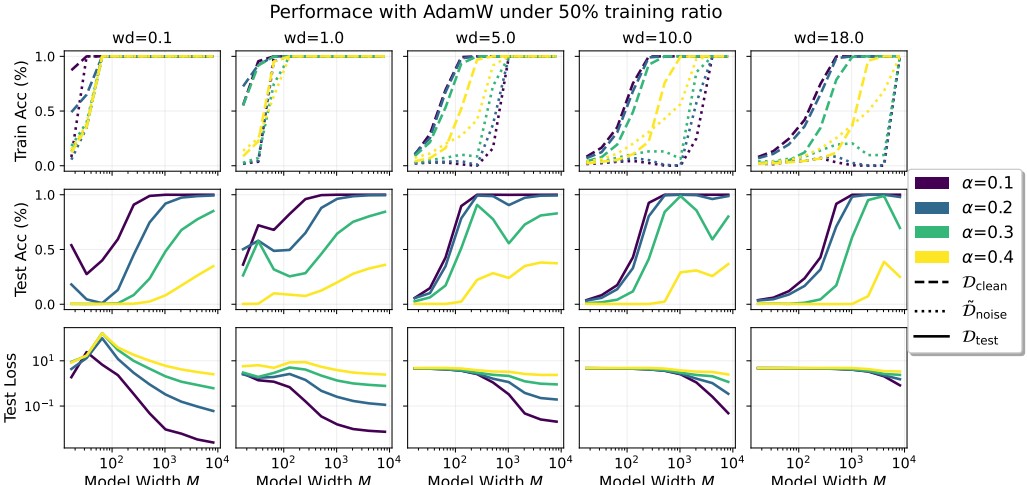

*Figure 3.* Performance of ReLU models trained with AdamW under varying weight decay. The first row shows training accuracy on clean ($\mathcal{D}_{\text{clean}}$) and noisy ($\tilde{\mathcal{D}}_{\text{noise}}$) data to characterize memorization. The second row shows test accuracy, and the third row shows test cross-entropy loss. Increasing weight decay shifts the interpolation threshold to larger model sizes, delaying the second-descent regime.

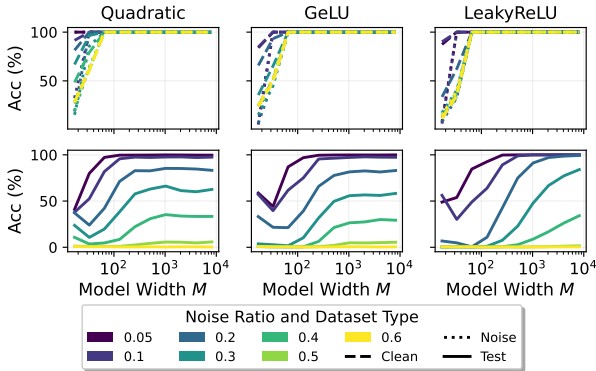

*Figure 4.* Performance across various activation functions. Generally, test accuracy is non-decreasing in the over-parameterized regime, but the ceilings of test accuracy exist for the quadratic and GeLU activations.

**Remarks on model misspecification.** We note that some model architectures are unable to interpolate all noisy labels, in which case the over-parameterized regime does not strictly exist. For example, models with tied first-layer weights ($\boldsymbol{U} = \boldsymbol{V}$) cannot memorize asymmetric random noise where $a \circ b \neq b \circ a$. In such cases, test accuracy does not vary monotonically with model size, and selecting an appropriate regularization strength may be more effective than increasing model capacity. We provide a detailed discussion in Section C.3.

### 3.2. Training Dynamics

We observe a counter-intuitive phenomenon in the training dynamics: noisy labels are consistently memorized faster than clean labels, as summarized in Result 3.2. While the

learning rates for these two components are not drastically different, this inversion of the typical learning order is robust across various optimizers, including AdamW, Adam, and Muon (see Section C.2 for details). Even for SGD, which generally requires significantly longer timescales to achieve generalization, this trend remains consistent (Figure 21).

**Result 3.2.** *In both under- and over-parameterized regimes, the model prioritizes the memorization of noisy labels over the fitting of clean labels.*

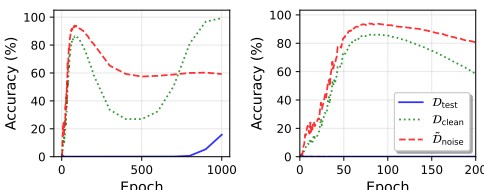

*Figure 5.* Typical learning curves in the under-parameterized regime. (*Left*) Trained on $\alpha = 0.3$ noisy data, model width $M = 2049$, using Adam optimizer with weight decay 0.001. (*Right*) A zoom-in of the left panel, showing the first 200.

Selected learning curves are shown in Figures 5 and 6. From the over-parameterized case in Figure 6, we observe two key patterns: (i) increased model capacity accelerates the memorization process, aligning with empirical findings in large language models (Tirumala et al., 2022); and (ii) the gap between the memorization speeds of noise and clean data becomes more pronounced at lower noise ratios $\alpha$.

This observation stands in contrast to existing theoretical studies suggesting that models initially fit correct samples and ignore noise, thereby making early stopping effective (Li et al., 2020). Our findings indicate that modular arith-

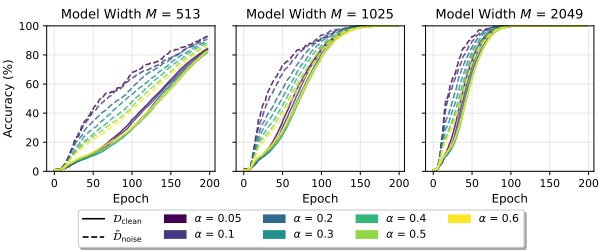

*Figure 6.* Learning curves in the over-parameterized regime. Training accuracy on noisy data rises faster than on clean data, indicating faster memorization of noise. Models are trained with AdamW.

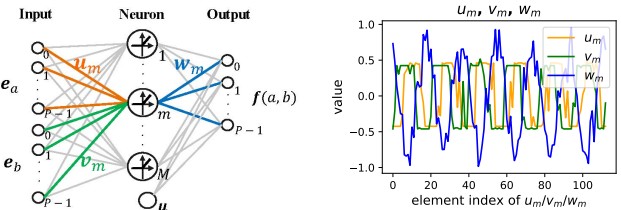

*Figure 7.* *(Left)* Illustration of a neuron's parameters $(\boldsymbol{u}_m, \boldsymbol{v}_m, \boldsymbol{w}_m)$. *(Right)* Visualization of a neuron from a ReLU model, with x-axis showing indices $0, \ldots, P - 1$.

metic tasks may exhibit unique dynamical properties, highlighting the need for further theoretical investigation into the mechanisms of label fitting in these settings. Additional results, including test accuracy learning curves, are provided in Section C.2.

# 4. Representation of Generalization

To understand how over-parameterized models resolve the tension between generalization and memorization, we investigate the internal representations learned by two-layer ReLU NNs. While exact solutions for modular addition is well-documented for models with quadratic activations (Pearce et al., 2023), the mechanism for untied ReLU models—especially under label noise—remains unclear.

**Analytical solution under quadratic activation.** The two-layer NN (1) can be equivalently written as the summation of the neurons:

$$\boldsymbol{f}_\theta\left(a, b\right) = \sum_{m \in \mathcal{M}} \boldsymbol{w}_m \phi\left(\boldsymbol{u}_m^\top \boldsymbol{e}_a + \boldsymbol{v}_m^\top \boldsymbol{e}_b\right) + \boldsymbol{\mu},$$

where $\mathcal{M} = \{1, \cdots, M\}$. The triple, $(\boldsymbol{u}_m, \boldsymbol{v}_m, \boldsymbol{w}_m)$, represents the $m^{\text{th}}$ neuron. Here, $\boldsymbol{w}_m$ is the $m^{\text{th}}$ column vector of $\boldsymbol{W}$, and $\boldsymbol{u}_m^\top$ and $\boldsymbol{v}_m^\top$ are the $m^{\text{th}}$ row vectors of $\boldsymbol{U}$ and $\boldsymbol{V}$, respectively. Figure 7 visualizes this decomposition.

In the case of quadratic activation ($\phi\left(\cdot\right) = \left(\cdot\right)^2$) for the modular addition task with prime $P > 2$, the NN model achieves $100\%$ generalization by taking the following ana-

lytical solution (Gromov, 2023; Morwani et al., 2024):

$$u_{mi} = \lambda \cos\left(\frac{2\pi}{P}\omega_m i + \varphi_m^{(a)}\right), \quad (2a)$$

$$v_{mj} = \lambda \cos\left(\frac{2\pi}{P}\omega_m j + \varphi_m^{(b)}\right), \quad (2b)$$

$$w_{mk} = \lambda \cos\left(\frac{2\pi}{P}\omega_m k + \varphi_m^{(c)}\right), \quad (2c)$$

where $\lambda \in \mathbb{R}$ is a scaling constant. The phases satisfy

$$\varphi_m^{(a)} + \varphi_m^{(b)} = \varphi_m^{(c)} + 2q\pi \quad (3a)$$

for some $q \in \mathbb{Z}$, and $\omega_m \in \left\{1, \cdots, \frac{P-1}{2}\right\}$. With sufficiently large width, every frequency $\omega \in \left\{1, \cdots, \frac{P-1}{2}\right\}$ has at least one neuron to cover, and empirically the distribution of the frequency is close to uniform. We present their theory (Morwani et al., 2024) in Section B.

For ReLU models, an exact analytical solution is not available. In this work, we find that ReLU models coarsely satisfy the necessary conditions of the analytical solution described above, except that the magnitude $\lambda$ of $w_{mk}$ differs from that of $u_{mi}$ and $v_{mj}$.

**Result 4.1.** *The generalization representation of ReLU models closely matches the analytical solution* (2)–(3) *for modular addition and subtraction tasks.*

In the remainder of this section, we provide empirical evidence for Result 4.1 through three key approaches. First, we replace the ReLU with the quadratic activation after training to examine the compatibility. Second, we apply neuron-wise frequency filtration to decouple generalization signals from memorization noise, thereby evaluating their separability. Finally, we validate the theoretical phase relationships defined in Equations (2)-(3) and examine the uniformity of the learned frequency distribution.

### 4.1. Evidence 1: Activation Replacement After Training

Notice that the analytical solution of quadratic activation is symmetric to the input in the sign of activation: in Equation (2), the periodicity of a neuron can be phase-shifted by $\pi$: $-\cos(A) - \cos(B) = \cos(A + \pi) + \cos(B + \pi)$, and $A + \pi + B + \pi = C + 2(q+1)\pi$ still matches (3). We also consider the Reverse ReLU defined as $\phi(x) = \max\{-x, 0\}$ (Figure 8).

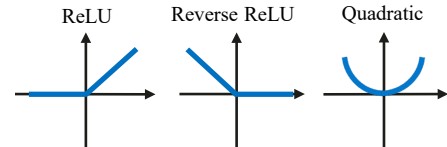

*Figure 8.* Visualization of the activation functions.

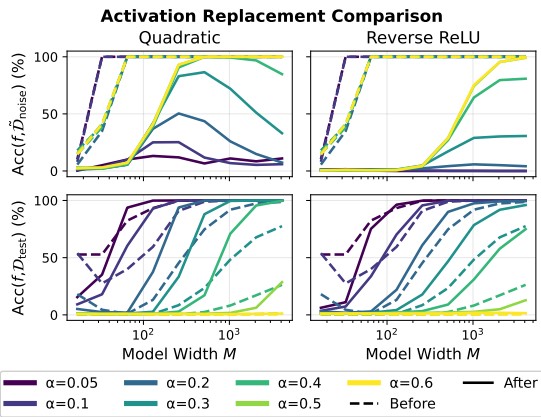

**Figure 9.** Performance before (dashed) and after (solid) replacing the ReLU with other activations after training. *(Lower)* Test accuracy improves after the replacement. *(Upper)* The memorization of the noise drops after replacement.

**Result 4.2.** *Replacing ReLU with quadratic/reverse ReLU activation after training not only preserves but even enhances generalization across modular addition and subtraction tasks.*

As illustrated in Figure 9, this activation replacement leads to a non-negligible increase in test accuracy. More importantly, it induces a label-correction effect: the model's accuracy on noisy training samples decreases because the corrupted labels are effectively flipped back to the ground-truth rules. The fact that the model maintains strong generalization under this replacement suggests that the learned weights $(\boldsymbol{u}_m, \boldsymbol{v}_m)$ and phases $\varphi_m$ form a symmetric representation capable of encoding the underlying rules.

### 4.2. Evidence 2: Dominant Frequency Filtration

Drawing inspiration from the analytical solutions of quadratic models, we hypothesize that the generalization signal in a ReLU neuron is carried by its dominant frequency.

**Neuron-wise Frequency Filtration (FF).** We perform a Fourier-based decomposition on each hidden neuron $m \in \mathcal{M}$, characterized by the parameters $(\boldsymbol{u}_m, \boldsymbol{v}_m, \boldsymbol{w}_m)$. Specifically, we apply a filter to isolate the frequency $\omega$ with the maximum magnitude in the weight vector $\boldsymbol{w}_m$, and retrieve the same frequency component from $\boldsymbol{u}_m$ and $\boldsymbol{v}_m$. This dominant component is referred to as the generalization part, $(\boldsymbol{u}_m^G, \boldsymbol{v}_m^G, \boldsymbol{w}_m^G)$, while the remaining frequency components constitute the residual part, $(\boldsymbol{u}_m^R, \boldsymbol{v}_m^R, \boldsymbol{w}_m^R)$. The formal mathematical definitions of this filtration process are detailed in Appendix D.1. We subsequently construct two sub-networks: one composed of the dominant frequency components, denoted by $\{\boldsymbol{U}^G, \boldsymbol{V}^G, \boldsymbol{W}^G, \boldsymbol{\mu}\}$, and another consisting of the residuals, $\{\boldsymbol{U}^R, \boldsymbol{V}^R, \boldsymbol{W}^R, \boldsymbol{\mu}\}$. By evalu-

ating these partitioned models separately, we quantify their respective contributions to memorization and generalization.

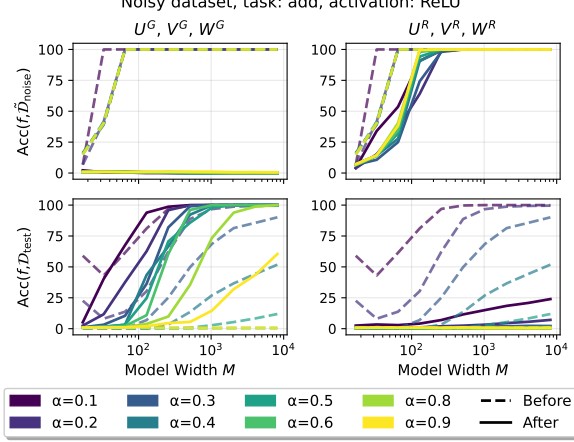

**Figure 10.** Performance before (dashed) and after (solid) FF for ReLU networks. The upper and lower panels show the accuracy on memorization ($\tilde{\mathcal{D}}_{\text{noise}}$) and generalization ($\mathcal{D}_{\text{test}}$) tasks, respectively. The filtered sub-network $\{\boldsymbol{U}^G, \boldsymbol{V}^G, \boldsymbol{W}^G, \boldsymbol{\mu}\}$ captures the structural generalization, whereas the residual component $\{\boldsymbol{U}^R, \boldsymbol{V}^R, \boldsymbol{W}^R, \boldsymbol{\mu}\}$ retains the noise memorization capability.

**Result 4.3.** *Sub-networks retrieved by frequency filtration (FF) largely disentangle generalization from memorization. The extracted sub-network $\{\boldsymbol{U}^G, \boldsymbol{V}^G, \boldsymbol{W}^G, \boldsymbol{\mu}\}$ is responsible for generalization and recovers strong test performance under severe label noise, whereas the complementary sub-network $\{\boldsymbol{U}^R, \boldsymbol{V}^R, \boldsymbol{W}^R, \boldsymbol{\mu}\}$ primarily memorizes noisy labels with little or no generalization capability. (Figure 10)*

The effect of frequency filtration (FF) is remarkable. The extracted sub-network $\{\boldsymbol{U}^G, \boldsymbol{V}^G, \boldsymbol{W}^G, \boldsymbol{\mu}\}$ recovers strong test performance under severe label noise, indicating that the underlying rules are already encoded in the model. In contrast, the complementary sub-network $\{\boldsymbol{U}^R, \boldsymbol{V}^R, \boldsymbol{W}^R, \boldsymbol{\mu}\}$ primarily memorizes noisy labels without contributing to generalization. Similar behaviors are observed on the modular subtraction task and for models with quadratic activations (see Section D.2). These results demonstrate a clear spectral disentanglement: dominant frequencies encode generalizable structure, while residual frequencies are primarily responsible for fitting label noise. Moreover, the findings suggest that the expression of the correct rules is suppressed during training by additional frequency components introduced to memorize noisy labels.

### 4.3. Evidence 3: Validation on Phases and Frequency

We validate the phase relationship and frequency distribution of the filtered dominant frequency ($U^G$, $V^G$, $W^G$) against the analytical solution (Equation (2)-(3)):

- **Phase Coherence:** For modular addition, the filtered

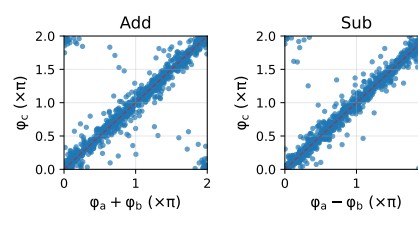

*Figure 11.* Relationship of phases under $M = 1025$ and $\alpha = 0.3$. Each point represents a neuron, where $\varphi_a$, $\varphi_b$, and $\varphi_c$ are the phases of $\boldsymbol{u}^G$, $\boldsymbol{v}^G$, and $\boldsymbol{w}^G$, respectively. The left is the modular addition task while the right corresponds to modular subtraction.

phases satisfy the circular relation Equation (3). Crucially, for the modular subtraction task, we identify the corresponding relation $\varphi_m^{(a)} - \varphi_m^{(b)} = \varphi_m^{(c)} + 2q\pi$ for $q \in \mathbb{Z}$ (Figure 11), proving that the ReLU model adapts its internal trigonometry to the specific group operation.

- **Spectral Uniformity:** We find that as model width $M$ increases, the distribution of dominant frequencies $\omega \in \{1, \ldots, \frac{P-1}{2}\}$ becomes increasingly uniform (see Figures in Section D.2). This suggests that scaling allows the model to cover the entire spectral representation of the task more densely, maybe a prerequisite for robust generalization.

The above evidences suggest that FF can disentangle memorization from generalization. However, FF is inherently limited in two respects: (i) it does not yield a strict functional decomposition, since $\boldsymbol{f}_\theta \neq \boldsymbol{f}_{\{\boldsymbol{W}^G, \boldsymbol{U}^G, \boldsymbol{V}^G, \boldsymbol{\mu}\}} + \boldsymbol{f}_{\{\boldsymbol{W}^R, \boldsymbol{U}^R, \boldsymbol{V}^R, \boldsymbol{\mu}\}}$; and (ii) it relies on prior knowledge of task-specific structure. This motivates the question of whether memorization and generalization can instead be separated in a task-agnostic manner.

## 5. Network Partitioning

We investigate whether a trained network can be partitioned into two disjoint neuron subsets: one subset $\mathcal{M}^G \subseteq \mathcal{M}$ responsible for generalization, and another subset $\mathcal{M}^R \subseteq \mathcal{M}$ responsible for memorizing label noise. Selecting $\mathcal{M}^G$ for generalization is equivalent to *pruning* the remaining neurons $\mathcal{M} \backslash \mathcal{M}^R$ from the full network. To this end, we propose Algorithm 1, which ranks neurons using an importance metric and selects subsets according to their contribution to generalization and noise memorization. Specifically, the resulting sub-networks can be written as

$$\boldsymbol{f}_{\mathcal{M}^G}(a, b) = \sum_{m \in \mathcal{M}^G} \boldsymbol{w}_m \phi\big(\boldsymbol{u}_m^\top \boldsymbol{e}_a + \boldsymbol{v}_m^\top \boldsymbol{e}_b\big) + \boldsymbol{\mu}, \quad (4a)$$

$$\boldsymbol{f}_{\mathcal{M}^R}(a, b) = \sum_{m \in \mathcal{M}^R} \boldsymbol{w}_m \phi\big(\boldsymbol{u}_m^\top \boldsymbol{e}_a + \boldsymbol{v}_m^\top \boldsymbol{e}_b\big) + \boldsymbol{\mu}. \quad (4b)$$

---

**Algorithm 1** Neuron Selection for Network Decomposition

1: **Input:** Network $\boldsymbol{f}_\theta$ with $M$ neurons $\mathcal{M}$; Metric $S(m)$; Val set $\mathcal{D}_{\text{val}}$; Noisy set $\hat{\mathcal{D}}_{\text{noise}}$; Threshold $\tau$.
2: **Initialize:** $\mathcal{M}^G \leftarrow \emptyset$, $\mathcal{M}^R \leftarrow \emptyset$.
3: {// Step 1: Ranking neurons}
4: Calculate score $s_m = S(m)$ for each $m \in \mathcal{M}$.
5: Sort neurons descending: $\{m_{\pi(1)}, \ldots, m_{\pi(M)}\}$ where $s_{m_{\pi(i)}} \geq s_{m_{\pi(j)}}$ for $i < j$.
6: {// Step 2: Selection for generalization}
7: **for** $k \in \{1, \ldots, M\}$ **do**
8:     $\mathcal{M}_k = \{m_{\pi(1)}, \ldots, m_{\pi(k)}\}$.
9:     $A_k = \text{Acc}(\boldsymbol{f}_{\mathcal{M}_k}, \mathcal{D}_{\text{val}})$.
10: **end for**
11: $k^G = \arg\max_k A_k$.
12: $\gamma^G = k^G/M$ and $\mathcal{M}^G = \{m_{\pi(1)}, \ldots, m_{\pi(k^G)}\}$.
13: {// Step 3: Minimal selection for noise memorization}
14: **for** $k \in \{1, \ldots, M\}$ **do**
15:     $\mathcal{M}'_k = \{m_{\pi(M-k+1)}, \ldots, m_{\pi(M)}\}$. {Bottom-up}
16:     **if** $\text{Acc}(\boldsymbol{f}_{\mathcal{M}'_k}, \tilde{\mathcal{D}}_{\text{noise}}) \geq \tau$ **then**
17:         $k^R = k$, **break**.
18:     **end if**
19: **end for**
20: $\gamma^R = k^R/M$ and $\mathcal{M}^R = \{m_{\pi(M-k^R+1)}, \ldots, m_{\pi(M)}\}$.
21: **Output:** Ratios $\gamma^G, \gamma^R$; Selected neurons $\mathcal{M}^G, \mathcal{M}^R$; Sub-networks $\boldsymbol{f}^G = \boldsymbol{f}_{\mathcal{M}^G}$, $\boldsymbol{f}^R = \boldsymbol{f}_{\mathcal{M}^R}$.

---

A practical importance scoring metric $S(m)$ is the *inverse participation ratio (IPR)* (Pastor-Satorras & Castellano, 2016; Girvin & Yang, 2019; Gromov, 2023; Doshi et al., 2024), proposed and applied in Doshi et al. (2024), measuring the periodicity of each neuron and defined as

$$\text{IPR}_m := \left(\frac{\|\tilde{\boldsymbol{w}}_m\|_4}{\|\tilde{\boldsymbol{w}}_m\|_2}\right)^4, \quad (5)$$

where $\|\cdot\|_P$ is the $l_p$-norm of the vector, and $\tilde{\boldsymbol{w}}_m$ is the Fourier transform of $\boldsymbol{w}_m$. Equation (5) indicates that $\text{IPR}_m \in \left[\frac{1}{M}, 1\right]$. The higher the value, the more periodic the vector. This metric is well-suited for modular addition and subtraction, which rely on periodic structures.

Alternatively, we propose a *task-agnostic* scoring metric, the *neuron strength* (abbreviated as Str.), defined as

$$\text{Str}_m := \|\boldsymbol{w}_m\|_\infty \phi\left(\|\boldsymbol{u}_m\|_\infty + \|\boldsymbol{v}_m\|_\infty\right), \quad (6)$$

where $\|\cdot\|_\infty$ denotes the $l_\infty$-norm.

Actually, Str. and IPR are highly correlated, and the task-agnostic property of Str. enables applying Algorithm 1 to the modular multiplication task, where periodicity is no longer the proxy for generalization. We formally state this observation in Result 5.1 and defer details to Section E.

**Result 5.1.** *Neurons that contribute most to generalization typically exhibit higher neuron strength. Specifically, neurons with high IPR also attain high Str. in the modular addition and subtraction tasks. (Figure 12 and 34)*

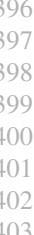
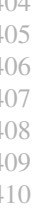
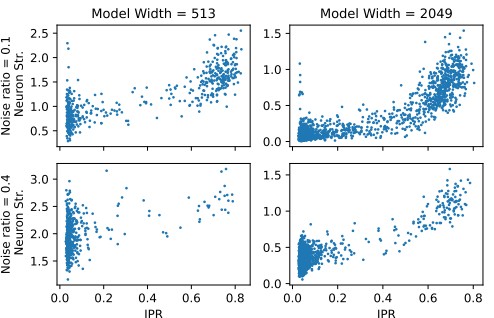

*Figure 12.* Neuron's IPR vs. Str. on the modular addition task.

We also observe from Figure 12 that a larger noise ratio reduces the number of high-IPR or high-Str. neurons, which may explain the performance deterioration under heavy noise. This trend is further supported by the optimal neuron selection ratios output by Algorithm 1 (see Figure 13).

**Result 5.2.** *The sub-networks are not overlapped when model width is sufficiently large, i.e., $\gamma^G + \gamma^R \leq 1$, where $\gamma^G = \frac{|\mathcal{M}^G|}{|\mathcal{M}|}$ and $\gamma^R = \frac{|\mathcal{M}^R|}{|\mathcal{M}|}$. (Figure 13)*

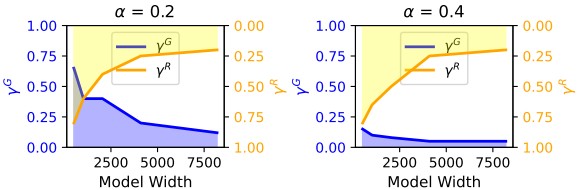

*Figure 13.* Neuron selection ratios using Str. on the modular addition task. Higher noise ratio leads to a smaller sub-network for generalization ($\gamma^G$). Two sub-networks have no overlapped neurons under sufficiently large model width.

**Result 5.3.** *Generalization improvement from pruning is limited compared with FF. (Figure 14)*

**Discussion on the entanglement of generalization and memorization.** Several conclusions can be drawn from Figure 14. First, neuron selection based on Str. yields performance comparable to that based on IPR, verifying the claim in Result 5.1. Second, neuron selection leads to noticeable generalization gains under light label noise ($\alpha \leq 0.4$). Third, although applying FF to the extracted sub-network $f^G$ further improves generalization, the gain remains substantially smaller than that achieved by applying FF directly to the full model (Result 5.3).

Together, these observations point to a fundamental entanglement between generalizable structure and noise mem-

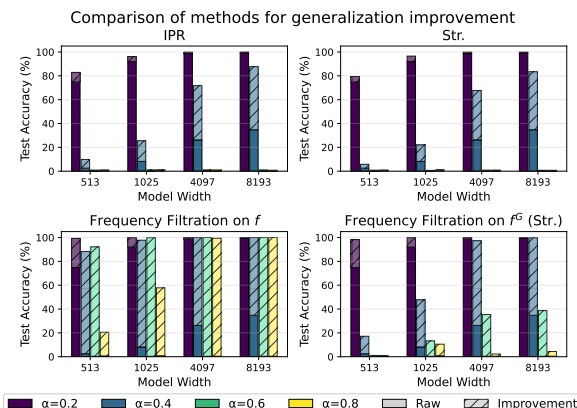

*Figure 14.* Performance of using IPR or Str. is close. FF on the sub-network $f^G$ retrieved by Str. further improves the generalization, but is still not comparable to FF on the raw model $f$. Performance is reported on the modular addition task.

orization in over-parameterized models. In particular, the limited benefit of post-hoc neuron selection suggests that generalizable structure is not localized to a small subset of neurons, but is instead diffusely encoded and intertwined with memorization. This leads to two important implications: (i) *Learning design:* future work should explore architectures or training paradigms that explicitly isolate memorization from logical inference during training (e.g., Deepseek Engram (Cheng et al., 2026)); (ii) *Knowledge extraction:* our results highlight the necessity of leveraging domain-specific structural priors, rather than task-agnostic neuron importance, to fully recover latent generalization capabilities that are otherwise suppressed by the requirement to interpolate noise.

**Conclusion.** This study investigates the entanglement between generalization and memorization in the over-parameterized model by the case of label noise. Models learn the underlying rules internally and distribute them across neurons, so retrieving the generalization sub-network by selecting/pruning neurons does not fully unleash the generalization capability. This suggests the need for the design of a learning paradigm that can naturally separate the memorization and generalization modules, and the creation of domain-specific extraction tools to fully retrieve the latent knowledge after training. In addition, the counter-intuitive observation that noisy labels are memorized faster than clean data presents an intriguing paradox for current learning theory. In conclusion, this work highlights fundamental limitations of post-hoc extraction and suggests new directions for designing learning paradigms with more explicit function seperation. These insights open several avenues for future research on representation learning and training dynamics in modern neural models.

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

# A. Related Work

**Model-wise Double Descent and Over-parameterization.** The discovery of the double descent phenomenon marked a significant shift in modern machine learning, suggesting that the classical bias–variance trade-off is incomplete (Belkin et al., 2019; Yang et al., 2020). In the over-parameterized regime, test error can decrease again after the interpolation threshold, leading to superior generalization. Theoretically, this second descent has been rigorously characterized in linear regression under various assumptions (Belkin et al., 2020; Advani et al., 2020; Adlam & Pennington, 2020; Bartlett et al., 2020; Mei & Montanari, 2022; Hastie et al., 2022; Muthukumar et al., 2020), although some studies indicate that appropriate regularization can mitigate or eliminate this effect (Nakkiran et al., 2020). In non-linear settings, double descent has been robustly observed in computer vision tasks, particularly in the presence of label noise (Nakkiran et al., 2021; Umar et al., 2025).

Recent efforts have sought mechanistic explanations for the generalization power of large-scale models. One perspective proposes that over-parameterized models generalize through smoother interpolation of noisy data points (Gamba et al., 2022; Somepalli et al., 2022). Furthermore, the emergence of neural scaling laws has been increasingly attributed to the superposition of discrete features or representations (Liu et al., 2025). Davies et al. (2023); Huang et al. (2024) further unify double descent with grokking by framing both phenomena as a competition between memorization and generalization circuits. Our work extends this line of research by investigating how model size specifically governs the resolution of this competition under label noise.

**Grokking and Emergent Generalization.** In over-parameterized models, generalization often emerges long after the training loss has vanished, a phenomenon known as grokking (Power et al., 2022). This decoupling between fitting and rule discovery is prominently observed in modular arithmetic tasks across various architectures (Liu et al., 2022b; Nanda et al., 2023; Mallinar et al., 2025). While specific algorithms, such as Pizza and Clock, have been identified for Transformers solving these tasks (Nanda et al., 2023; Zhong et al., 2023), and a universal abstract algorithm for deep neural networks has been uncovered (McCracken et al., 2025), the exact analytical solution for two-layer ReLU networks remains unsolved. Existing theoretical works have explored provable solutions using quadratic activations or specific regularization schemes (Gromov, 2023; Tian, 2025), yet the role and impact of label noise in these settings remain largely unexplored.

**Learning under Label Noise.** Most existing work on learning with label noise focuses on robust training strategies to prevent overfitting (Li et al., 2020; Liu et al., 2022a). We refer readers to Song et al. (2022) for a comprehensive survey. Xu et al. (2023) theoretically analyze the performance of two-layer ReLU networks on noisy XOR data under the assumption of fixed second-layer weights, which deviates from standard end-to-end training dynamics. Doshi et al. (2024) examine the effects of noise ratios and different regularization methods on arithmetic tasks. In contrast, we primarily focus on the relationship between model size and generalization performance, and further investigate the mechanistic entanglement between memorization and generalization.

# B. Analytical Solution Under Quadratic Activation

The fact that the solution in Equations (2) and (3) is exact for quadratic activations has been well discussed in Pearce et al. (2023); Morwani et al. (2024). We present their theory here, adapted to our notation, in Proposition B.1.

**Proposition B.1.** *(Theorem 7, Morwani et al. (2024)) For the modular addition task, any two-layer network $f_\theta = W \left( U e_a + V e_b \right)^2$ with $M \geq 4(P-1)$ that maximizes the margin on $\mathcal{D}_{total}$ satisfies the following conditions:*

*1. for each neuron $\{ \boldsymbol{u}_m, \boldsymbol{v}_m, \boldsymbol{w}_m \}$, there exists a scaling constant $\lambda \in \mathbb{R}$ and a frequency $\omega \in \left\{ 1, \cdots, \frac{P-1}{2} \right\}$, such that*

$$u_{mi} = \lambda \cos\left( \frac{2\pi}{P} \omega_m i + \varphi_m^{(a)} \right), \tag{7a}$$

$$v_{mj} = \lambda \cos\left( \frac{2\pi}{P} \omega_m j + \varphi_m^{(b)} \right), \tag{7b}$$

$$w_{mk} = \lambda \cos\left( \frac{2\pi}{P} \omega_m k + \varphi_m^{(c)} \right), \tag{7c}$$

*for some phase offsets $\varphi_m^{(a)}, \varphi_m^{(b)}, \varphi_m^{(c)} \in \mathbb{R}$ satisfying $\varphi_m^{(a)} + \varphi_m^{(b)} = \varphi_m^{(c)}$.*

*2. For every frequency $\omega \in \left\{ 1, \cdots, \frac{P-1}{2} \right\}$, at least one neuron in the network uses this frequency.*

# C. Supplementary Experiments for Section 3

## C.1. Double-descent curves on varying setups

Model-wise double descent is clearly observed across different activation functions, training ratios, optimizers, and arithmetic tasks. We summarize the Figures about it in Table 1.

*Table 1.* Figure indices showing double descent curves under varying setups.

| Activation Functions | Training Ratios | Optimizers | Arithmetic Tasks |
|:---:|:---:|:---:|:---:|
| 4, 15 | 16 (40%), 3(50%), 17(60%) | 3 (AdamW), 18 (Adam), 19 (Muon) | 20 |

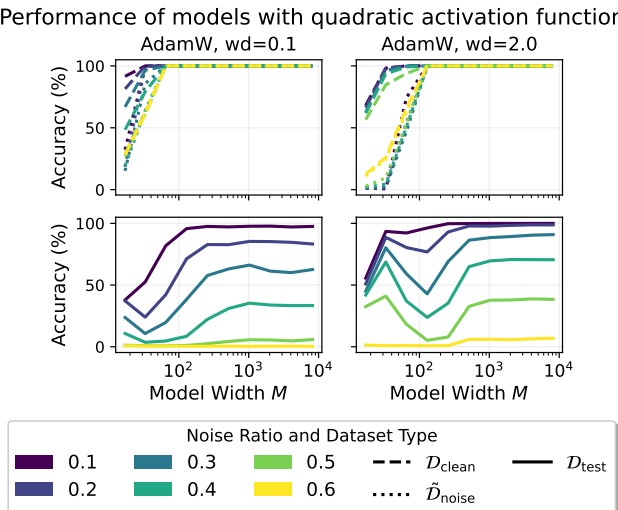

*Figure 15.* Ceilings of test accuracy for quadratic activation functions exist across varying weight decays.

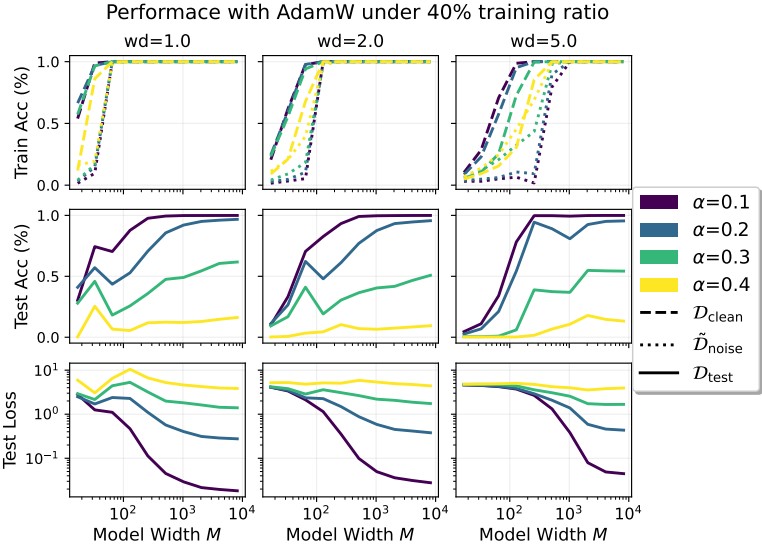

*Figure 16.* Performance of AdamW under 40% training ratio.

**SGD is not a good choice.** Figure 21 shows the training dynamics of SGD under different choices of learning rate and weight decay. Among all configurations, only the setting with lr = 0.1 and wd = 0.001 exhibits noticeable generalization behavior. This suggests that SGD requires highly restrictive hyperparameter tuning to achieve reasonable generalization.

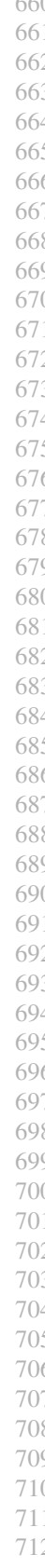

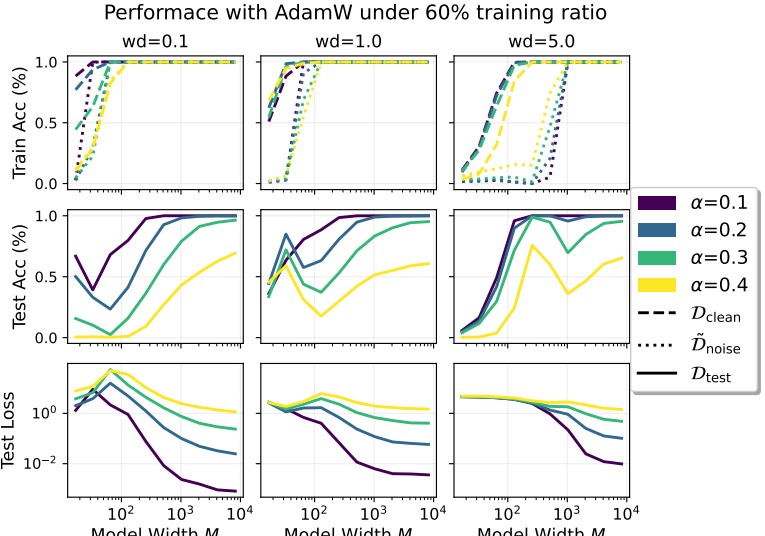

*Figure 17.* Performance of AdamW under 60% training ratio.

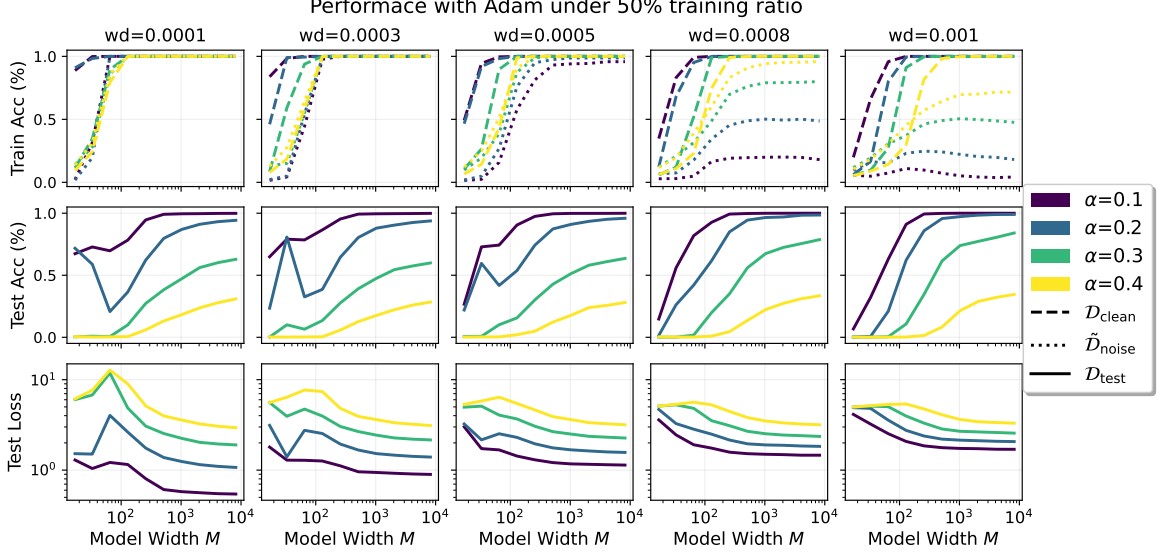

*Figure 18.* Performance of Adam under varying weight decays.

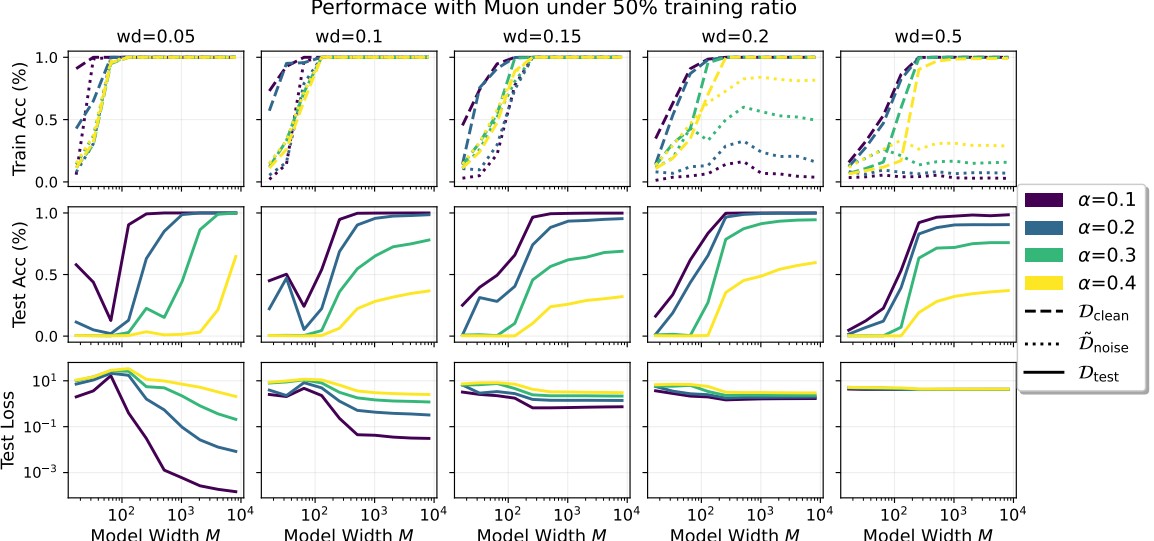

*Figure 19.* Performance of Muon under varying weight decays.

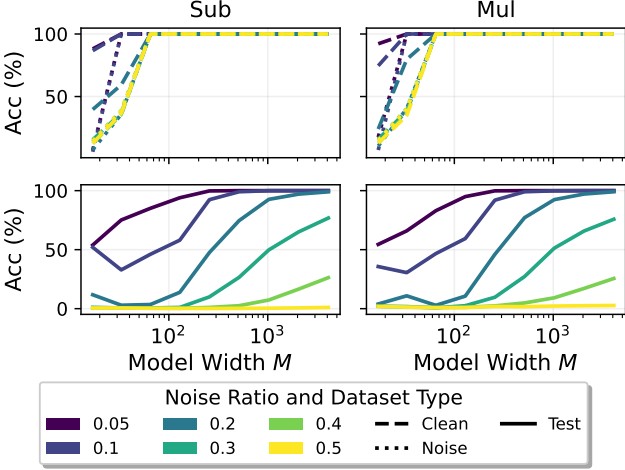

*Figure 20.* Performance on modular subtraction (*left*) and multiplication (*right*) tasks. Models are trained using AdamW under 50% training ratio.

Moreover, even at epoch 200,000, the test accuracy remains relatively low and is still increasing, indicating that SGD converges extremely slowly. Overall, SGD is inefficient for this task, as it consumes an excessive number of training epochs to reach a modest level of generalization.

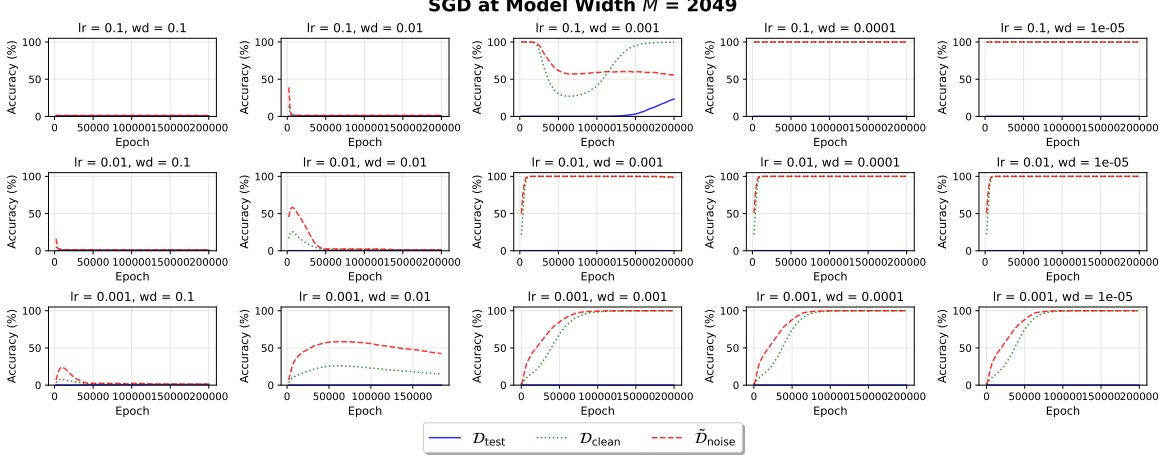

*Figure 21.* Training dynamics of SGD under different learning rates (lr) and weight decay (wd) values at $M = 2049$ and $\alpha = 0.3$. Results are reported every 2,000 epochs, where the total epochs are 200,000.

## C.2. Accuracy learning curves

The accuracy learning curves capture when memorization and generalization emerge during training. We observe the grokking phenomenon even under noisy labels: the model quickly reaches 100% training accuracy, while test accuracy improves significantly only later. To illustrate this, we separately present the training dynamics on short and long time scales, highlighting memorization and generalization, respectively.

### C.2.1. MEMORIZATION ON A SHORT TIME SCALE

Without noise in the training dataset, previous studies have found that larger models memorize the training data faster (Tirumala et al., 2022; Huang et al., 2024). This trend persists under noisy training data, although memorization speeds differ for clean and noisy subsets. We find that the model memorizes noisy labels faster than clean labels, and that the gap is larger at lower noise ratios. These phenomena are consistently observed across several optimizers, including Adam (Figure 22), AdamW (Figure 23), and Muon (Figure 24).

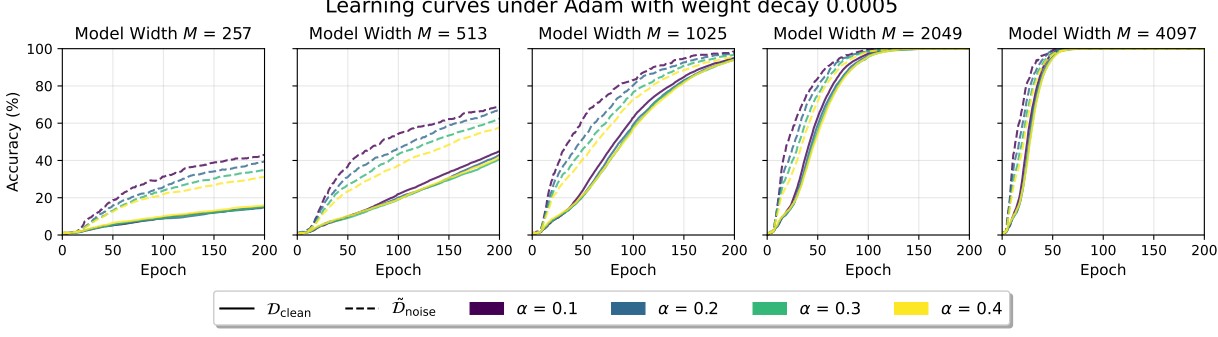

*Figure 22.* Memorization curves for the Adam optimizer. Larger models reach high training accuracy more quickly. Noisy labels are memorized faster than clean labels, and lower noise ratios further accelerate this effect.

### C.2.2. GENERALIZATION ON A LONG TIME SCALE

In terms of test accuracy, we observe a general tendency that models under larger weight decay grok faster, although the trend is not equally pronounced across all hyperparameter settings. This trend is clear for AdamW (Figure 25) and Muon

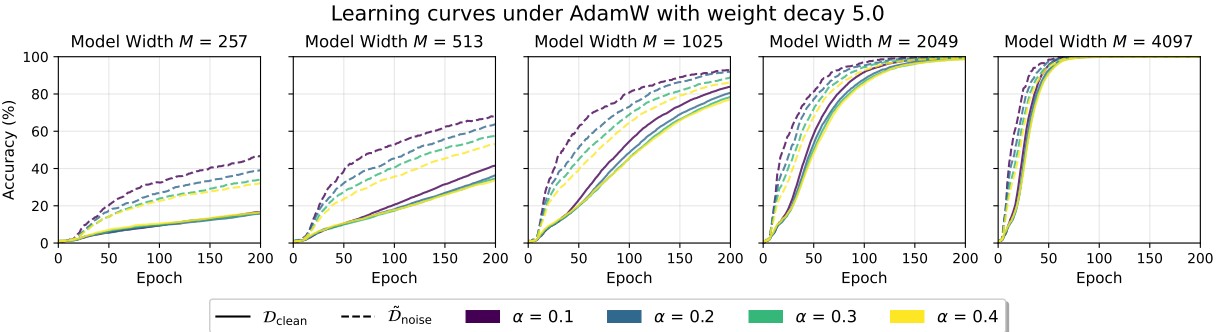

*Figure 23.* Memorization curves for AdamW optimizer.

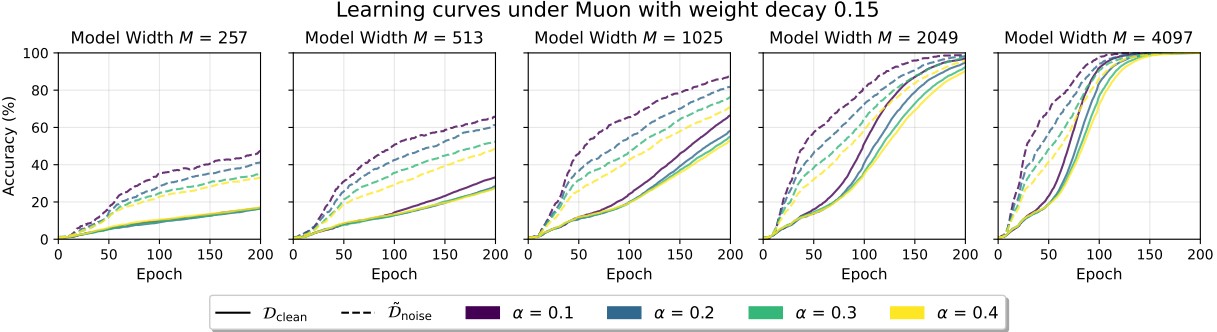

*Figure 24.* Memorization curves for Muon optimizer.

(Figure 27), but not evident for Adam (Figure 26). Across all of them, models under smaller noise ratio grok faster.

### C.3. Discussion on Model Misspecification

We consider a scenario where the model architecture is not expressive enough to memorize all noisy labels, which leads to atypical generalization behavior.

Modular addition and multiplication are commutative operations, i.e., $a \circ b = b \circ a$. Accordingly, an alternative architecture enforces a tied first layer by setting $U = V$ (Pearce et al., 2023). Under random label noise, however, the noisy samples generally violate this symmetry. As a result, the tied model is unable to interpolate all noisy labels, and the coexistence phase does not emerge in this setting, leading to irregular test accuracy curves (see the left panel in Figure 28).

To verify that the asymmetric structure of random noise is responsible for this behavior, we construct a symmetric noise pattern and re-evaluate the tied model. Specifically, for a noise ratio $\alpha$ in modular addition, we randomly sample $\frac{\alpha}{2}|\mathcal{D}_{\text{train}}|$ examples to form $\mathcal{D}_{\text{asym}}$, where each $(a, b, c) \in \mathcal{D}_{\text{asym}}$ satisfies $a < b$. The noisy dataset is then constructed as

$$\tilde{\mathcal{D}}_{\text{noise}} = \{(a, b, \tilde{c}) \mid (a, b, c) \in \mathcal{D}_{\text{asym}} \text{ or } (b, a, c) \in \mathcal{D}_{\text{asym}}, \ \tilde{c} \neq c, \ \tilde{c} \in \{0, \ldots, P-1\}\}.$$

Under this symmetric noise, the tied model exhibits predictable behavior: noisy labels can be fully memorized once the model width exceeds a threshold (i.e., the coexistence phase appears), and generalization improves with increasing model size (see the middle panel in Figure 28).

When the noise structure is unknown, model misspecification becomes a potential risk. One practical mitigation strategy is to strengthen regularization so as to extend the inversion phase. The tied model is able to interpolate all clean samples, and increasing weight decay encourages the model to suppress/neglect noisy labels, thereby recovering good generalization performance (see the right panel in Figure 28).

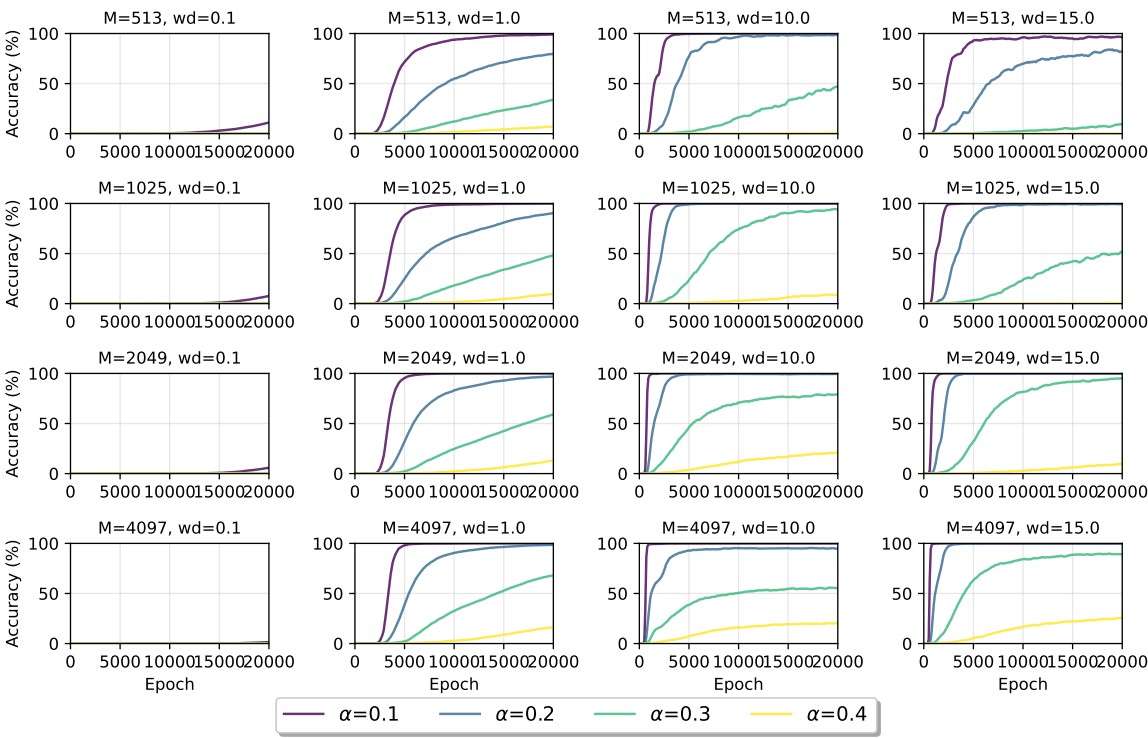

*Figure 25.* Training dynamics of test accuracy across various weight decays for AdamW. Small weight decay (wd=0.1) leads to delayed and slow grokking.

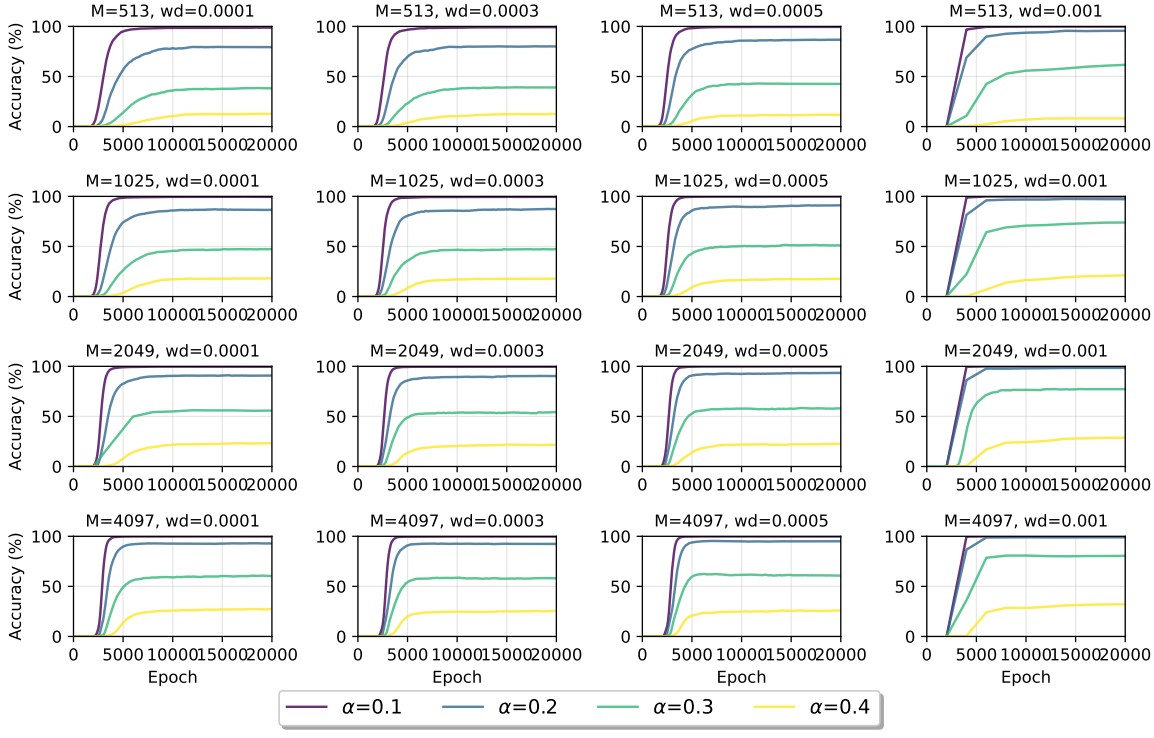

*Figure 26.* Training dynamics of test accuracy across various weight decays for Adam.

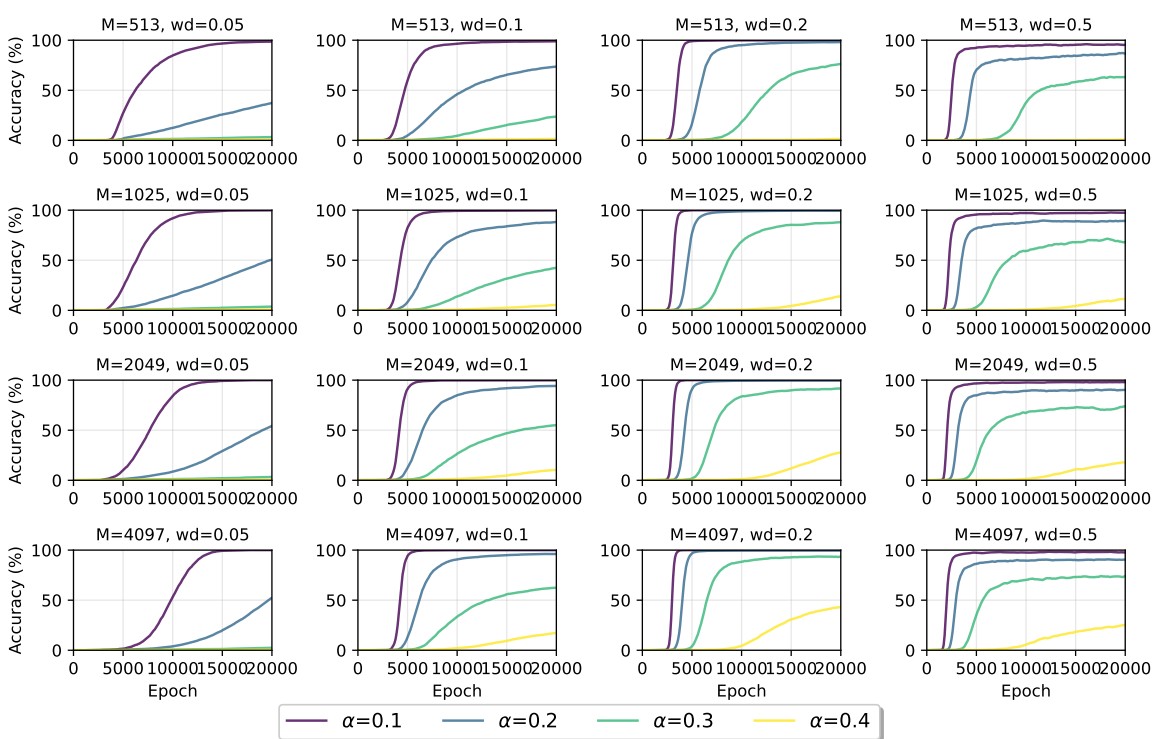

*Figure 27.* Training dynamics of test accuracy across various weight decays for Muon.

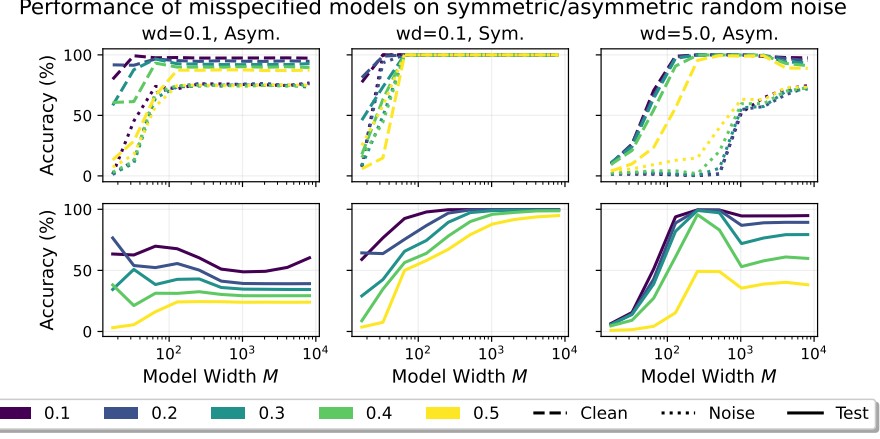

*Figure 28.* Performance of first-layer tied models under model misspecification. *(Left)* With asymmetric random noise, the tied model fails to fully interpolate noisy labels and exhibits irregular generalization behavior. *(Middle)* Under symmetric noise, the tied model successfully interpolates all noisy labels, and generalization improves with increasing model size. *(Right)* Stronger regularization mitigates the adverse effect of misspecification by encouraging the model to ignore noisy labels.

# D. Supplementary Materials for Section 4

## D.1. Details of frequency filtration

In this section, we provide more details about the frequency filtration described in Section 4.2. That is to say, the details about decomposing $(\boldsymbol{u}_m, \boldsymbol{v}_m, \boldsymbol{w}_m)$, the $m^{\text{th}}$ neuron, into $\left(\boldsymbol{u}_m^G, \boldsymbol{v}_m^G, \boldsymbol{w}_m^G\right)$ and $\left(\boldsymbol{u}_m^R, \boldsymbol{v}_m^R, \boldsymbol{w}_m^R\right)$.

Let $\omega = e^{-2\pi i/P}$. Define the Fourier transform matrix $F \in \mathbb{C}^{P \times P}$ by

$$F_{k,n} = \frac{1}{\sqrt{P}} \omega^{kn} = \frac{1}{\sqrt{P}} e^{-2\pi i k n/P}.$$

For any vector $\boldsymbol{x} \in \mathbb{R}^{P \times 1}$, denote its Fourier transform as $\tilde{\boldsymbol{x}} = \boldsymbol{F}\boldsymbol{x}$. $\boldsymbol{F}$ is a unitary matrix, so its conjugate transpose of $\boldsymbol{F}$, denoted as $\boldsymbol{F}^H$, gives $\boldsymbol{F}^H \boldsymbol{F} \boldsymbol{v} = \boldsymbol{v}$. Without loss of generality, we assume $\boldsymbol{x}$ is a vector satisfying that $\arg \max_k |\tilde{x}_k|$ is a singleton.

We define the main frequency retrieval operator (for generalization part) as $\boldsymbol{F}^H \boldsymbol{F}_G(\boldsymbol{x})$, filtering the frequency with maximum magnitude, namely,

$$F_G(\boldsymbol{x})_{k,n} = \begin{cases} F_{k,n}, & k \in \arg \max_{k'} |\tilde{x}_{k'}| \\ 0, & k \notin \arg \max_{k'} |\tilde{x}_{k'}| \end{cases},$$

and the remaining frequency retrieval operator (for memorization) as $\boldsymbol{F}^H \boldsymbol{F}_R(\boldsymbol{x})$, where $\boldsymbol{F}_R(\boldsymbol{x}) = \boldsymbol{F} - \boldsymbol{F}_G(\boldsymbol{x})$.

For the $m^{\text{th}}$ neuron, let $\boldsymbol{P}_{G,m} = \boldsymbol{F}^H \boldsymbol{F}_G(\boldsymbol{w}_m)$ and $\boldsymbol{P}_{R,m} = \boldsymbol{F}^H \boldsymbol{F}_R(\boldsymbol{w}_m)$. Then, we define the decomposition to neuron $m$ as

$$\boldsymbol{u}_m^G = \text{Real}(\boldsymbol{P}_{G,m}\boldsymbol{u}_m), \ \boldsymbol{u}_m^R = \text{Real}(\boldsymbol{P}_{R,m}\boldsymbol{u}_m),$$
$$\boldsymbol{v}_m^G = \text{Real}(\boldsymbol{P}_{G,m}\boldsymbol{v}_m), \ \boldsymbol{v}_m^R = \text{Real}(\boldsymbol{P}_{R,m}\boldsymbol{v}_m),$$
$$\boldsymbol{w}_m^G = \text{Real}(\boldsymbol{P}_{G,m}\boldsymbol{w}_m), \ \boldsymbol{w}_m^R = \text{Real}(\boldsymbol{P}_{R,m}\boldsymbol{w}_m).$$

It is easy to check $\boldsymbol{P}_{G,m} + \boldsymbol{P}_{R,m} = \boldsymbol{I}$, so

$$(\boldsymbol{u}_m, \boldsymbol{v}_m, \boldsymbol{w}_m) = \left(\boldsymbol{u}_m^G, \boldsymbol{v}_m^G, \boldsymbol{w}_m^G\right) + \left(\boldsymbol{u}_m^R, \boldsymbol{v}_m^R, \boldsymbol{w}_m^R\right).$$

We provide the visualization of a neuron under such frequency filtration in Figure 29.

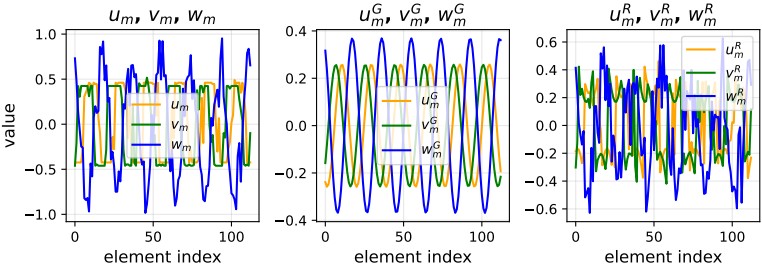

*Figure 29.* Visualization of applying frequency filtration to a neuron. $\left(\boldsymbol{u}_m^G, \boldsymbol{v}_m^G, \boldsymbol{w}_m^G\right)$ retains the frequency component with the maximum magnitude in $(\boldsymbol{u}_m, \boldsymbol{v}_m, \boldsymbol{w}_m)$.

## D.2. Frequency filtration benefits generalization under varying setups

Frequency filtration also disentangles generalization and memorization for the modular *subtraction* task (Figure 30) and for models with *quadratic* activations (Figure 31). We observe substantial improvement in generalization after frequency filtration: ReLU models recover test accuracy under severe label noise, while quadratic models exceed previously observed accuracy limits. Test accuracy also increases for models trained on noise-free data (Figure 32).

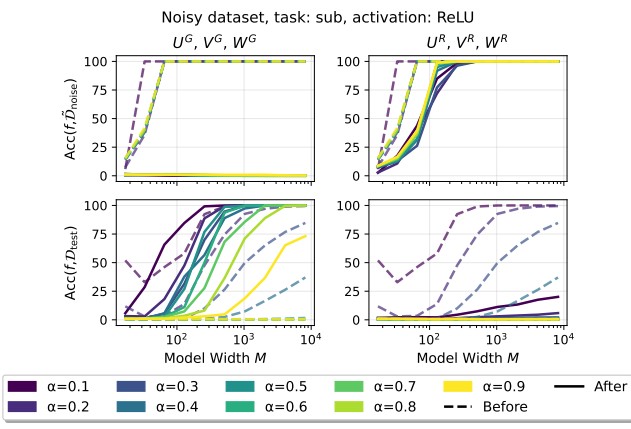

*Figure 30.* Performance before (dashed) and after (solid) the frequency filtration on the modular *subtraction* task.

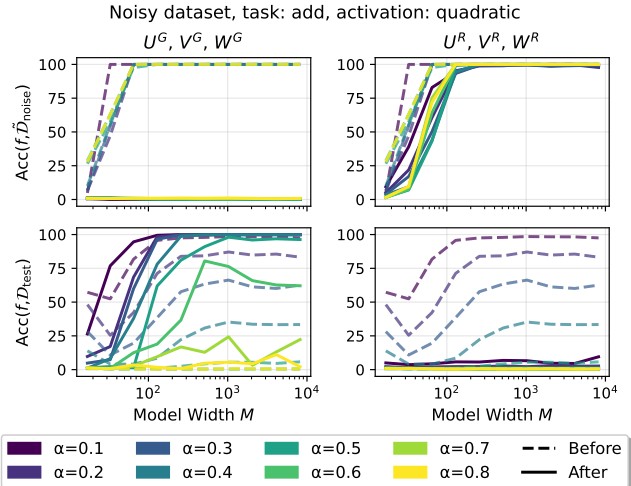

*Figure 31.* Performance before (dashed) and after (solid) the frequency filtration on models with *quadratic* activation. We remark that frequency filtration also breaks the ceilings of test accuracy.

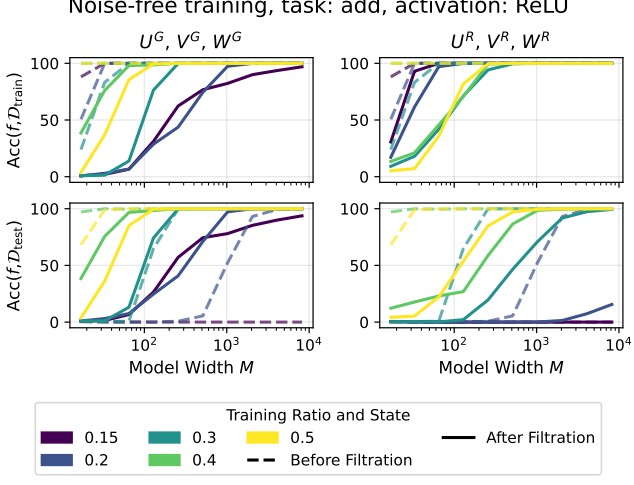

*Figure 32.* Performance before (dashed) and after (solid) frequency filtration for models trained on the clean modular addition data. The models use ReLU activation and are trained with AdamW. Overall, test accuracy after filtration ($U^G$, $V^G$, $W^G$) is generally higher, particularly for large models.

## D.3. Relationship among the phases $\varphi_m^{(a)}$, $\varphi_m^{(b)}$, and $\varphi_m^{(c)}$

Suppose the dominant frequency of neuron $m$ has the form

$$u_{mi}^G = \lambda_u \cos\left(\frac{2\pi}{P}\omega_m i + \varphi_{m,G}^{(a)}\right), \tag{8a}$$

$$v_{mj}^G = \lambda_v \cos\left(\frac{2\pi}{P}\omega_m j + \varphi_{m,G}^{(b)}\right), \tag{8b}$$

$$w_{mk}^G = \lambda_w \cos\left(\frac{2\pi}{P}\omega_m k + \varphi_{m,G}^{(c)}\right). \tag{8c}$$

For models trained with either ReLU or quadratic activations, there exists $q \in \mathbb{Z}$ such that the following equations are approximately satisfied:

$$\varphi_{m,G}^{(a)} + \varphi_{m,G}^{(b)} = \varphi_{m,G}^{(c)} + 2q\pi \qquad \text{(modular addition)}, \tag{9}$$

$$\varphi_{m,G}^{(a)} - \varphi_{m,G}^{(b)} = \varphi_{m,G}^{(c)} + 2q\pi \qquad \text{(modular subtraction)}. \tag{10}$$

We quantify the empirical deviation of these relationships using the mean squared error (MSE). Specifically, for modular addition:

$$\text{Phase MSE} = \frac{1}{M}\sum_{m=1}^{M} \min_{q\in\mathbb{Z}} \left(\varphi_{m,G}^{(a)} + \varphi_{m,G}^{(b)} - \varphi_{m,G}^{(c)} + 2q\pi\right)^2,$$

and for modular subtraction, the $+$ is replaced by $-$ in the definition above.

We visualize these relationships in Figures 33 (addition) and 34 (subtraction), and the Phase MSE trends with respect to model size and noise ratio are shown in Figure 35. We observe that higher noise ratios generally lead to larger Phase MSE, i.e., larger deviation to the analytical relationship.

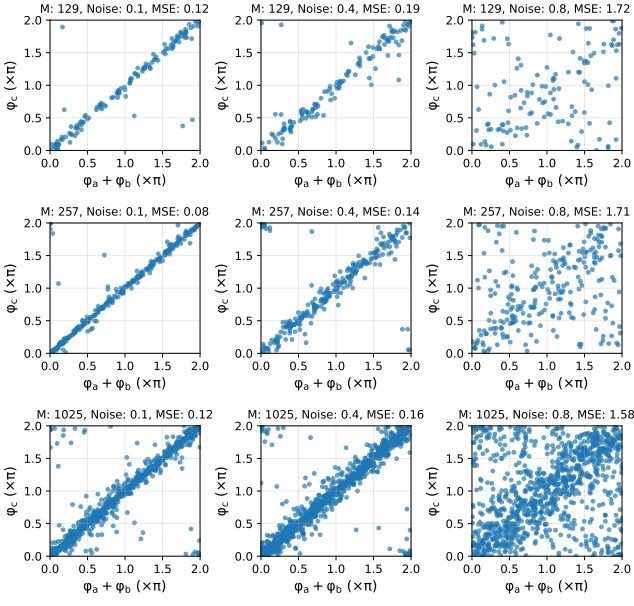

*Figure 33.* Scatter plot showing the relationship of phases on modular addition tasks. Each point represents a neuron, where $\varphi_a$, $\varphi_b$, and $\varphi_c$ are the phases of $\boldsymbol{u}^G$, $\boldsymbol{v}^G$, and $\boldsymbol{w}^G$, respectively.

## D.4. Uniform distribution of the frequency

Figure 36 shows that each frequency is covered by at least one neuron after frequency filtration (only dominant frequency left for each neuron). Figure 37 also shows that the distribution of the frequency magnitude after filtration is close to uniform

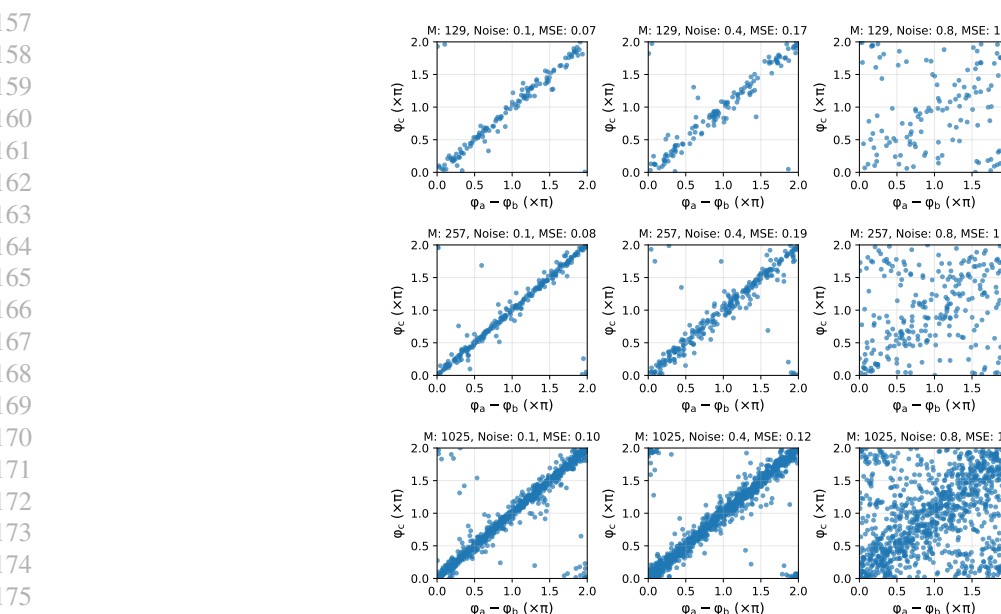

*Figure 34.* Scatter plot showing the relationship of phases on modular subtraction tasks. Each point represents a neuron, where $\varphi_a$, $\varphi_b$, and $\varphi_c$ are the phases of $\boldsymbol{u}^G$, $\boldsymbol{v}^G$, and $\boldsymbol{w}^G$, respectively.

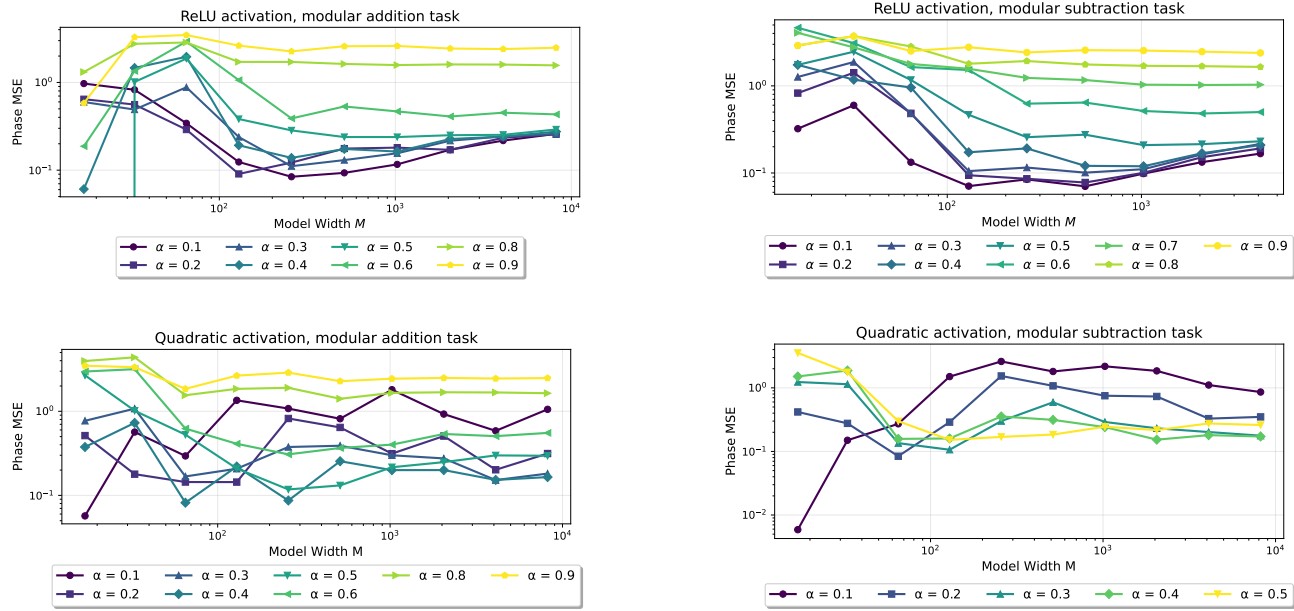

*Figure 35.* Phase MSE across different tasks, activation functions, model widths, and noise ratios. Higher noise ratios increase Phase MSE for ReLU models, and a similar trend is observed for quadratic models on the modular addition task.

at larger model width. Here, let $\tilde{W}^G$ is the Fourier transform of $W^G$ with each column constituted by $\{w_m\}_{m=1}^M$, so Figure 37 plots the norm of each row vector in $\tilde{W}^G$ with index in $\left\{1, \cdots, \frac{P-1}{2} + 1\right\}$.

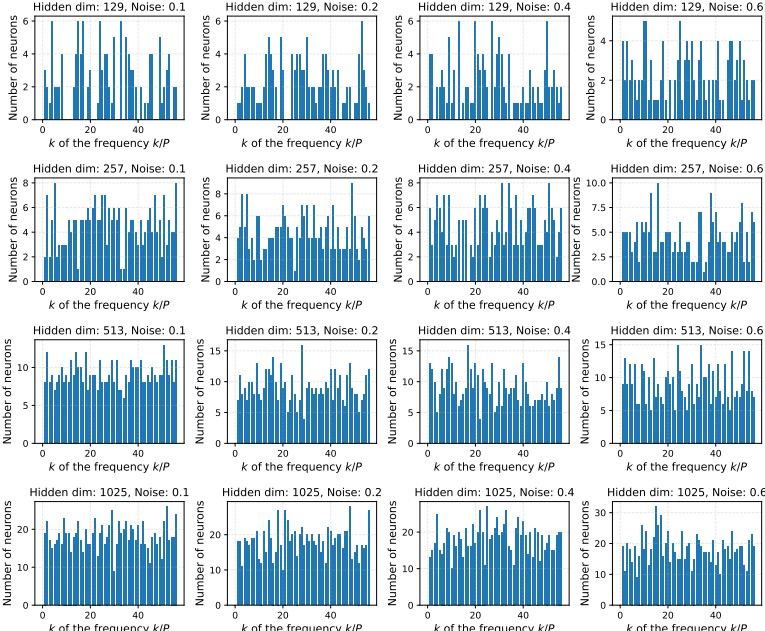

*Figure 36.* Each frequency $\omega \in \left\{1, \cdots, \frac{P-1}{2}\right\}$ is covered by at least a neuron when the model width is no less than 513.

## E. Supplementary Materials for Section 5

### E.1. Performance of Algorithm 1 using Str. or IPR

In this section, we examine generalization improvements across different tasks achieved through neuron selection. Figure 38 reports the improvement obtained using IPR for neuron selection, while Figure 39 presents the corresponding results for Str.

Comparing the results on the modular multiplication task illustrates the generality of the method using Str. This task lacks periodic patterns relevant for generalization, so measuring neuron importance via IPR offers limited improvement.

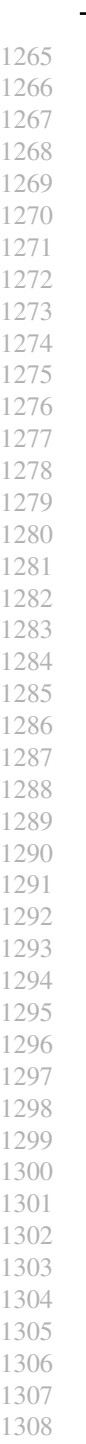
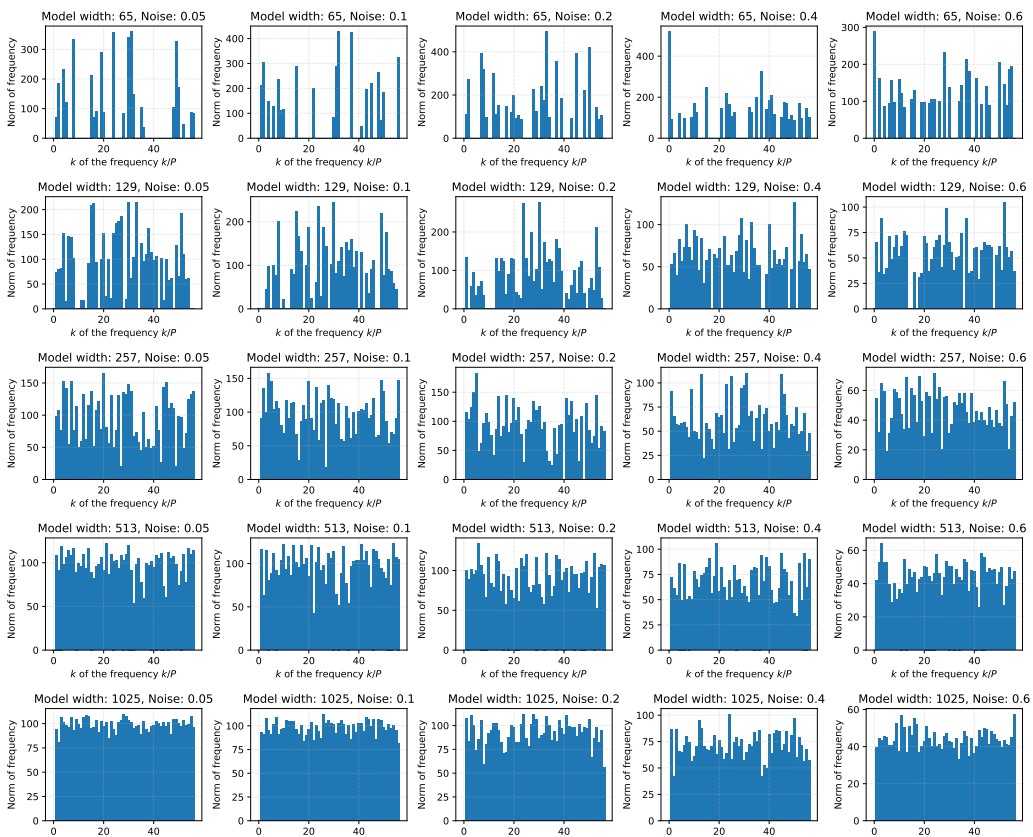

*Figure 37.* The magnitude (norm) of each frequency $\omega \in \left\{ 1, \cdots, \frac{P-1}{2} \right\}$ of $\boldsymbol{W}^G$ after frequency filtration to each neuron.

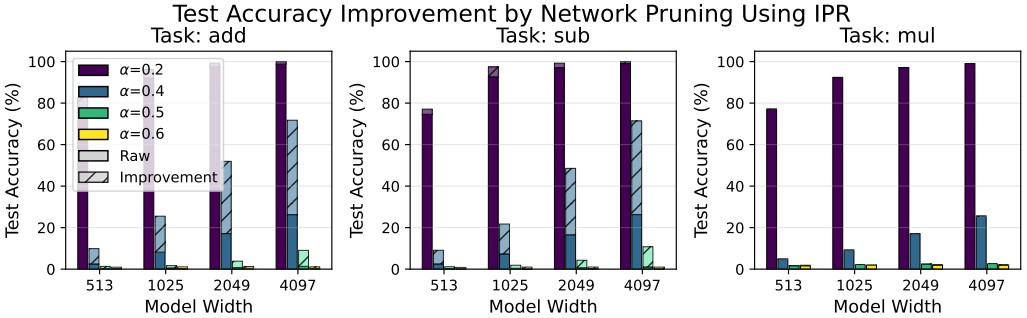

*Figure 38.* Generalization improvement across various tasks by neuron selection/pruning using IPR. No noticeable improvement is observed for the modular multiplication task.

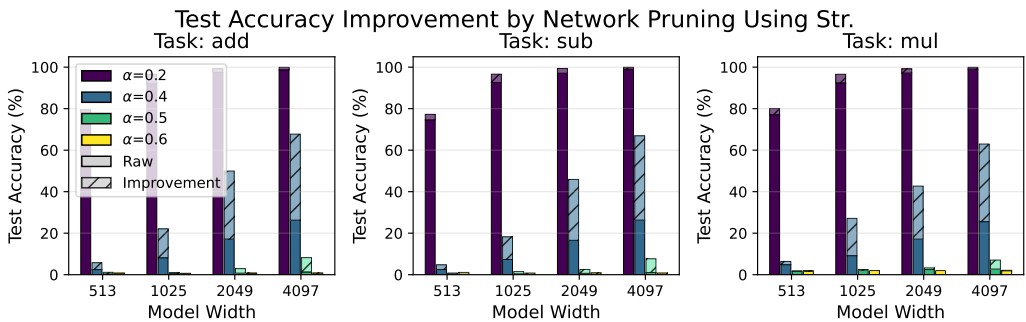

*Figure 39.* Generalization improvement across various tasks by neuron selection/pruning using Str. Significant improvement is also observed for the modular multiplication task.

