# OpenReview forum: "Unveiling the Coexistence of Generalization and Memorization: A Case Study on Arithmetic Tasks with Label Noise"
_ICML.cc/2026/Conference — Submitted to ICML 2026_

### Official Review · Reviewer_hjyW · 2026-02-26

**Soundness:** 3
**Presentation:** 2
**Significance:** 2
**Originality:** 3
**Overall Recommendation:** 3
**Confidence:** 3

**Summary:**

The paper studies the possibility of separating generalization and memorization under a constraint setting (i.e., modular arithmetic and a two-layer neural network). The authors try to answer several important questions, e.g., when label noise exists, whether a bigger model is better; even when the label noise ratio is extremely large, can the model still learn generalizable rules; is it possible to separate these two abilities from neuron perspectives, and so on.

The paper is self-contained, solid in both theory and experimental parts, but a bit dense to me. My main concern is that the applicability of the reported results could be extended to other, more practical settings, since many of the properties of the model’s behavior rely on the periodic nature of the modular arithmetic task. I find several methods are quite inspiring and have the potential to be extended to more general regression or classification tasks, e.g., the network partitioning in Section 5. Having discussions and results other than the modular arithmetic would significantly strengthen the paper and extend its applicability as well. Furthermore, a more formal discussion of these phenomena from the gradient or dynamical perspectives would also enhance the generalizability of the results here.

Hence, although I like the paper, I can only give a borderline rejection at this stage.

**Compliance With Llm Reviewing Policy:**

Affirmed.

**Key Questions For Authors:**

The problem studied in this paper (especially Figure 1) is closely related to benign overfitting (where noisy labels are perfectly interpolated without degrading generalization). What is the relationship between the results here and benign overfitting?

**Limitations:**

Yes

**Strengths And Weaknesses:**

## Strength:

- Well-controlled setup. The paper studies an important problem, i.e., separating memorization and generalization, under a very clean and controllable setting.
- Comprehensive empirical validation. The reported phenomena (double descent under noise, faster memorization of noisy labels, internal recovery of structural rules) are consistently validated across different optimizers, model widths, activation functions, and noise ratios.
- Insightful representation-level analysis. The frequency-based decomposition offers a compelling perspective on how generalization signals and memorization components coexist within the network. The dominant-frequency filtration provides evidence that rule structure is internally encoded even under extreme noise.

## Weakness:

- The paper reports a very counterintuitive phenomenon that the noisy label is learned faster than the clean labels, which is good. Although the authors provide sufficient experiments supporting this, it lacks a theoretical or dynamical analysis. Given the strong structural properties of modular arithmetic tasks (e.g., Fourier decomposability), a gradient-space or frequency-level analysis would significantly strengthen the claim, and hence clarify whether this phenomenon is task-specific or fundamental.
- Analyzing using the two-layer network is fine, but the modular arithmetic task would introduce some unique properties of the optimal representations, which might be hard to generalize to other tasks. Hence, experiments on more general tasks (e.g., classification, regression, etc.) would be helpful. For example, section 5 claims a task-agnostic decomposition framework via neuron strength (Str.) as an importance metric. It would substantially strengthen the paper to validate this decomposition beyond modular arithmetic.

---

> ### Author Rebuttal · Authors · 2026-03-30
>
> We thank the reviewer for the encouraging feedback and for finding our work *inspiring* and *insightful*. We are grateful that the reviewer appreciated our *well-controlled setup* and *comprehensive empirical validation*. To address the concerns regarding the generalizability of our findings and the need for a dynamical perspective, we have performed additional experiments and analysis as detailed below.
>
> ## Dicussion about the counterintuitive phenomenon
>
> We appreciate the reviewer for highlighting this counterintuitive phenomenon at the start of training. Our experiments show this behavior is not task-specific but depends on using full batch training.
>
> ### 1. Generalizability on MNIST
>
>
> We tested a three-layer network on MNIST with noise ratios (**$\alpha$**) of 0.05 and 0.2. In the first epoch, the accuracy of noisy samples actually exceeds that of clean samples.
>
> | **Period** | **α=0.05, noisy** | **α=0.05, clean** | **α=0.2, noisy** | **α=0.2, clean** |
> | ------------------ | -------------------------- | -------------------------- | ------------------------- | ------------------------- |
> | epoch 1          | 10.87%                   | 2.86%                    | 10.75%                  | 2.89%                   |
> | epoch 2          | 6.47%                    | 46.29%                   | 5.71%                   | 48.09%                  |
>
> This effect is fragile; it usually disappears if the batch size is significantly smaller than the full dataset.
>
> ---
>
> ### 2. Gradient analysis
>
> To understand why this happens, we look at how the loss changes. The total loss **$L$** is the sum of clean loss **$L_c$** and noisy loss **$L_n$**:
>
> $$
> L(\theta)=\frac1D\sum_{i\in S_c}\ell_i(\theta)+\frac1D\sum_{i\in S_n}\ell_i(\theta)
> = L_c(\theta)+L_n(\theta),
> $$
>
> where $D$ is the size of the training dataset, and $S_c$ and $S_n$ are the subsets of the clean and the noisy data.
>
> The learning direction (gradient) for each part is **$g_c$** and **$g_n$**, where the total direction is **$g = g_c + g_n$**. The speed at which each group learns is determined by how its own direction aligns with the total direction:
>
> $$
> -\frac{dL_c}{dt} = \langle g_c, g \rangle = \|g_c\|^2 + \langle g_c, g_n \rangle
> $$
>
> $$
> -\frac{dL_n}{dt} = \langle g_n, g \rangle = \|g_n\|^2 + \langle g_n, g_c \rangle
> $$
>
> We define the learning rate per sample as **$r_c$** and **$r_n$** by dividing by the number of samples in each group (**$D_c$** and **$D_n$**):
>
> $$
> r_c = \frac{1}{D_c} (\|g_c\|^2 + \langle g_c, g_n \rangle)
> $$
>
> $$
> r_n = \frac{1}{D_n} (\|g_n\|^2 + \langle g_c, g_n \rangle)
> $$
>
> At the very beginning, we observe that **$\langle g_c, g_n \rangle < 0$**, meaning the two groups pull in opposite directions. However, because the number of noisy samples **$D_n$** is much smaller than **$D_c$**, the ratio **$\|g_n\|^2 / D_n$** is much larger than **$\|g_c\|^2 / D_c$**.
>
> In simple terms, each noisy sample receives a much stronger individual 'push' from the gradient than each clean sample does. This makes the noisy group's accuracy climb faster initially. If we do not use full batches, this specific 'push' is scattered and inconsistent, which is why the effect is rarely seen in standard training. We will add more discussion about it in the next version.
>
> ## Limitations of modular arithmetic tasks
>
> We agree that testing beyond modular arithmetic is important. In the above, we have shown that the phenenomon result 3.2 can generalize to the MNIST dataset under certain conditions. For MNIST, we found that label noise is hard to memorize in a small model (with noise acc < 5%). When we applied the pruning method from Section 5 to it, both the test accuracy and the noisy label accuracy dropped together with the increase in the pruning ratio. This also supports our conclusion: NNs with the current training paradigm do not naturally separate noise from real patterns in their physical structure.
>
> The value of our modular arithmetic study lies in two areas. First, it allows us to study the grokking mechanism, which is still not fully understood for common network setups. Second, the frequency filtration in Section 4 provides a perfect benchmark to see exactly how much we can separate memory from generalization. The fact that pruning fails to separate them even in simpler tasks highlights that structural disentanglement is a difficult and fundamental challenge. We tend to leave a more general design of separation as future work.
>
> ## Connection to benign overfitting
>
> We thank for pointing out the connection to ​benign overfitting​. Our work connects to this field in two ways: (1) Most benign overfitting theories rely on linear or kernel assumptions. We study two-layer ReLU networks. (2) While benign overfitting literature focuses on the final error rate, we look at the internal representation. In words, benign overfitting discusses the conditions to generalize, and we complementarily discuss the inner working mechanism.

---

> > ### Author Rebuttal · Reviewer_hjyW · 2026-04-05
> >
> > Please refer to the comments below.

---

> > > ### Author Response · Authors · 2026-04-06
> > >
> > > Thank you for acknowledging our rebuttal. We are eager to hear your follow-up questions so that we can provide detailed responses and additional information as needed.

---

### Official Review · Reviewer_rDik · 2026-03-12

**Soundness:** 2
**Presentation:** 2
**Significance:** 2
**Originality:** 2
**Overall Recommendation:** 3
**Confidence:** 3

**Summary:**

This study probes potential mechanisms by which generalizations and memorizations coexist in neural networks trained on classification. Specifically, the authors trained a 2-layer network trained on modular arithmetic using clean and noisy labels and analyzed it. Their analyses are based on the fact that a 2-layer network can implement exact solutions and that noisy labels cannot be inferred from a rule but should be memorized. This means that correct prediction of noisy labels reflects memorization, whereas correct prediction of unseen examples reflects generalization.

**Compliance With Llm Reviewing Policy:**

Affirmed.

**Key Questions For Authors:**

1. Are D_{clean} and D_{noise} disjointed?

2. Can the authors explain clearly how the network maintains its accuracy after the activation functions are swapped? The results are surprising, but they should provide more meaningful insights regarding their results and their claim.

3. In frequency filtering, why do the authors use only *one* dominant frequency to find generalization signals? This seems somewhat unusual, since periodicity is used to partition the network (Eq. 5). 1.

**Limitations:**

No, the authors did not discuss the limitations, but I do not think this study has any negative impact.

**Strengths And Weaknesses:**

Strengths

This study addresses one of the fundamental questions in machine learning, which would be of great interest to the readers/audience of the conference. The authors provide some interesting observations: 1) the network can memorize noisy data faster than clean data, 2) units in the the hidden layer can be split into the generalization and memorization sub-networks.

Weaknesses

1. Concerns regarding the task and models:

1.1. The authors consider modular arithmetic task as a classification task and use it to better understand neural networks’ capacity to learn classification. However, modular arithmetic is not a traditional classification task because it can be solved by memorization.
1.2. The authors write, “We robustly observe the double descent of the test error across different activation functions (Figure4)” in lines 159-160. However, in some cases, the double descent is not pronounced at all. Even when the double descent is observed, the trend is very modest or very weakly pronounced. This suggests that the task is not suitable to study the double descent.
1.3. This study relies on a 2-layer network, which is not a common neural network architecture. It is neither MLP nor transformers. I recommend that the authors clarify the intended implications for MLP or transformers.

2. Concerns regarding clarification and/or presentation of their results

2.1. In section 3.1., the authors discussed the regularization but did not explicitly explain which regularization is used. Figure 3 implies that the weight decay is used as regularization, but it is unclear why the weight decay (among other regularizations) is specifically chosen to control model complexity and how it controls model complexity.

2.2. The authors write, “When the EMC is limited, learning signals for generalization and memorization compete more strongly. As a result, increase in regularization can be beneficial when memorization of label noise is undesirable”. This statement is extremely vague and needs to be further explained. They also need to explain why the limited capacity would favor generalization over memorization.

---

> ### Author Rebuttal · Authors · 2026-03-30
>
> We thank the reviewer for the feedback. We added experiments on Transformers and MNIST to show our findings apply to different models and tasks. Our responses are below.
>
> ## 1. The setup of the experiments
>
> > modular arithmetic is not a traditional classification task because it can be solved by memorization ...
> > Are D_{clean} and D_{noise} disjointed?
>
> The clean and noisy datasets are disjoint. This is why the model cannot solve the task simply by memorizing. The total available dataset only has $P\times P$ samples, so one can imagine it is a $P\times P$ table, where the training dataset fills part of it. The test dataset is corresponding to the blank of this table. Therefore, memorizing the training dataset is hard to generalize to the test dataset. That's why this task is popular in investigating the grokking phenomenon, where generalization occurs long after fully memorizing the training dataset to 100% accuracy.
>
> > ... 2-layer network... clarify the implications for MLP or transformers.
>
> We use a 2-layer network because it is a simple way to see how memorization and generalization happen at the same time. Also, this structure is a basic part of a Transformer. To show our results apply to other models, we ran experiments on a Transformer. By keeping only the top 20 frequencies in the last layer, we found that the model ignores noise and performs better on test data.
>
> | | clean acc | noisy acc | test acc|
> |---|---|---|---|
> | Before frequency filtration | 100% | 100% | 76.8% |
> | After frequency filtration | 98.1% | 36.8% | 87.8% |
>
> We also tested a 3-layer network on the MNIST dataset. We found that the model still memorizes noisy labels faster than clean ones under some settings. (More discussion about it can be found in the response to Reviewer hjyW):
>
> |  | $\alpha=0.05$, noisy | $\alpha=0.05$, clean | $\alpha=0.2$, noisy | $\alpha=0.2$, clean |
> | --- | --- | --- | --- | --- |
> | epoch 1 | 10.87% | 2.86% | 10.75% | 2.89% |
> | epoch 2 | 6.47% | 46.29% | 5.71% | 48.09%  |
>
> [1] Nanda, N., Chan, L., Lieberum, T., Smith, J., & Steinhardt, J. Progress measures for grokking via mechanistic interpretability. In ​*The Eleventh International Conference on Learning Representations*​.
>
> > ... explicitly explain which regularization is used ... it is unclear why the weight decay (among other regularizations) is specifically chosen to control model complexity and how it controls model complexity ... ... explain why the limited capacity would favor generalization over memorization
>
> We thank the reviewer for pointing out the need for clarification. We specifically chose weight decay (WD) as our regularization method because it is the most direct way to control the effective model complexity (EMC) by constraining the $L_2$ norm of the model parameters.
>
> Mathematically, Weight Decay modifies the loss function $\mathcal{L}_0$ as follows:
>
> $$
> \mathcal{L}(\theta) = \mathcal{L}_0(\theta) + \frac{\lambda}{2} \|\theta\|_2^2
> $$
>
> During gradient descent, the update rule becomes:
>
> $$
> \theta_{t+1} = (1 - \eta\lambda)\theta_t - \eta \nabla \mathcal{L}_0(\theta_t)
> $$
>
> where $\eta$ is the learning rate and $\lambda$ is the WD coefficient. By penalizing large weights, WD restricts the model to a smaller hypothesis space (a ball in parameter space with a smaller radius), so the EMC is low. This makes it harder for the model to memorize every noisy point and encourages it to find simpler, general rules instead. In [2], the authors give an example with low complexity (with only 5 neurons) to show the disappearance of the grokking phenomenon.
>
> [2] Pearce, A., Ghandeharioun, A., Hussein, N., Thain, N., Wattenberg, M., & Dixon, L. (2023). Do machine learning models memorize or generalize. ​*People+ AI Research*​.
>
> ## 2. Explanations of the results
>
> > ... in some cases, the double descent is not pronounced ...
>
> The first descent in Figure 4 is sometimes hard to see because our starting model size is already large enough to fit the data well. However, our study focuses on the second descent. This is where larger models perform better on test data even after they have fully memorized the training set, which is a key interest in modern machine learning.
>
> > Can the authors explain clearly how the network maintains its accuracy after the activation functions are swapped? The results are surprising, but they should provide more meaningful insights regarding their results and their claim. ... why do the authors use only one dominant frequency to find generalization signals? ...
>
> We are also surprised when we obtain this observation. Our results suggest the learned rule of this task does not rely on the exact activation shape. If the internal representation is symmetric, swapping functions will not break the logic. This shows the general rule is more important than the specific activation used. The reason for using single frequency is that one frequency shows the strongest periodicity (highest IPR in Equ 5).

---

> > ### Author Rebuttal · Reviewer_rDik · 2026-04-03
> >
> > Modular arithmetic has a finite number of inputs, and an input in each case has zero variability. For this type of task, one does not need to use any complex system other than memory. In contrast, for traditional classification tasks such as image classification, the variability is virtually infinite. Every individual cat looks different. I am not sure how results from problems with a finite number of inputs with zero variability can be generalized into problems with highly variable inputs.
> >
> > The authors stated “We use a 2-layer network because it is a simple way to see how memorization and generalization happen at the same time. Also, this structure is a basic part of a Transformer.” However, the fact that 2-layer network is the part of the transformer does not warrant that their results can be generalized to transformers.

---

> > > ### Author Response · Authors · 2026-04-06
> > >
> > > We sincerely thank the reviewer for the thoughtful feedback and for pointing out these considerations.
> > >
> > > Regarding the first concern, we agree that the input space in modular arithmetic is indeed finite. We chose this task for the following reasons: (1) It provides a clear and verifiable boundary between *memorization* and *generalization*. In this setting, a model is fully capable of achieving 100% training accuracy through pure memorization without generalizing, allowing us to rigorously test whether the model has truly learned the underlying rules. (2) The algorithms learned by models on *clean* data in this task are well-documented in existing literature, including studies on 2-layer NNs with quadratic activation [1], Transformers [2], and non-parametric methods [3]. Building on these established foundations, we further investigate how neural networks form internal algorithms specifically in the presence of label noise.
> > >
> > > Regarding the second concern, we acknowledge that the fact that an MLP is a component of a Transformer does not automatically guarantee that our findings will generalize to the entire architecture. To address this, we have provided additional results in the rebuttal by applying frequency filtration to the final head of a Transformer. The results demonstrate that this method also helps the model mitigate label noise and enhances its generalization capabilities. We will include a detailed discussion of these findings in the final manuscript.
> > >
> > > [1] *Feature emergence via margin maximization: case studies in algebraic tasks*, ICLR (Spotlight), 2024.
> > >
> > > [2] Nanda et al., *Progress measures for grokking via mechanistic interpretability*, ICLR (Spotlight), 2023.
> > >
> > > [3] Mallinar et al., *Emergence in non-neural models: grokking modular arithmetic via average gradient outer product*, ICML (Oral), 2025.

---

### Official Review · Reviewer_ahKx · 2026-03-12

**Soundness:** 3
**Presentation:** 3
**Significance:** 3
**Originality:** 3
**Overall Recommendation:** 4
**Confidence:** 3

**Summary:**

The authors propose an empirical study on the entanglement between memorization and generalization in the neural network learning process, focusing on a two-layer NN model and modular addition, subtraction, and multiplication tasks. Through their empirical evaluation process, the authors highlight a few different findings which might be relevant for the theoretical and complete understanding of how NNs learn and generalize from data. The most interesting findings seem to be relative to the quick memorization of noisy labels and the impossibility to fully disentangle memorization and generalization. The paper is interesting and quite easy to follow, while also being relevant for the ML community. However, a few questions and doubts persist, especially linked with the generalizability of the findings.

**Compliance With Llm Reviewing Policy:**

Affirmed.

**Final Justification:**

The authors provided a detailed rebuttal which addressed most of my concerns. My general opinion about the paper has not changed, still being more positive than negative. The paper tackles a relevant issue, highlighting interesting findings and is well written. My main concern is linked with the rather simple nature of the task considered throughout the experimental analysis. Therefore, I will keep my original score of weak accept.

**Key Questions For Authors:**

- The authors focus explicitly on a case study with a two-layer NN and a fairly small set of learning tasks that are quite simple (modular addition, subtraction, and multiplication). Therefore, I wonder how many of the results expressed by the authors could translate as they are to more complex settings, where the models are much larger and the task are much more complex. Could the authors provide some insights on this regard?
- The research questions Q1, Q2, Q3 are defined in the introduction and later on ignored in the remainder of the paper. The definition of such questions is relevant and defines a good frame for the paper topic. However, the authors should consider to reference such questions whenever they propose the answers for them.
- Why did the authors consider to use Muon as an optimizer? The usage of a fairly new optimizer such as Muon should be better justified. Also, would it be possible to extend the findings to models trained using a sharpness-aware minimization (SAM) optimizer? SAM and its variations [1,2] have been shown to be more generalizable while also being more susceptible to memorization [3,4]. Therefore, it would be interesting for the authors to consider if their findings apply to SAM.
- The two rows of plots in figure 4 refer to train and test accuracy respectively? The figure only mentions accuracy, so it is not completely clear. The authors should consider to clarify this aspect.
- The finding on noisy labels being consistently memorized faster than clean labels is in sharp contrast with the findings of papers that investigate the learning behaviour of atypical/noisy samples [5,6]. These papers have shown that atypical samples (which correlate to noisy labels or uncommon features) tend to be memorized more and later on during the training process of an over-parametrized model. Therefore, I tend to be a bit skeptical about the authors finding 3.2 and I wonder if this might be due to the specific experimental settings that the authors are considering. Could the authors provide a more in-depth discussion on how the memorization of noisy labels link to the memorization process of atypical samples to justify their findings?
- The results presented in section 4 and especially result 4.1 is a bit fuzzy and difficult to understand. Could the authors provide a clarification on the usefulness of the findings? Why would it matter to know that the generalization representation of ReLU models closely matches the analytical solution and how would such analytical solutions be used in practice? Also, the result 4.1 seems to apply only to modular addition and subtraction tasks. Therefore, I wonder what can be said for the multiplication task.
- The results on frequency filtration proposed in figure 10 are very interesting. Would it be possible to use them in practice to filter out or reduce the effect of memorization in complex neural networks? Since the investigation made by the authors apply only to two-layers NNs, I wonder how this filtration approach can be used in practical scenarios, where NNs are much larger.
- Is algorithm 1 and generally speaking the results of section 5 applicable to a larger NN model? If so, could the authors provide an example of the application of their study to a complex learning task such as image classification in computer vision or text processing with LLMs?

[1]. Kwon, Jungmin, et al. "Asam: Adaptive sharpness-aware minimization for scale-invariant learning of deep neural networks." International conference on machine learning. PMLR, 2021.

[2]. Li, Tao, et al. "Friendly sharpness-aware minimization." Proceedings of the IEEE/CVF conference on computer vision and pattern recognition. 2024.

[3]. Foret, Pierre, et al. "Sharpness-aware minimization for efficiently improving generalization." arXiv preprint arXiv:2010.01412 (2020).

[4]. Kim, Young In, et al. "On memorization and privacy risks of sharpness aware minimization." arXiv preprint arXiv:2310.00488 (2023).

[5]. Toneva, Mariya, et al. "An empirical study of example forgetting during deep neural network learning." arXiv preprint arXiv:1812.05159 (2018).

[6]. Agiollo, Andrea, Young In Kim, and Rajiv Khanna. "Approximating memorization using loss surface geometry for dataset pruning and summarization." Proceedings of the 30th ACM SIGKDD Conference on Knowledge Discovery and Data Mining. 2024.

**Limitations:**

The authors did not provide a discussion on the limitations of their work. Given the weaknesses and my questions, I would suggest the authors to add a brief discussion on if and how the results are general.

**Strengths And Weaknesses:**

STRENGHTS:
- Analyzing memorization and generalization is vital to fully understand the NN learning process
- The paper highlights several interesting findings about the memorization-generalization entanglement
- The experimental evaluation is extensive over the chosen setting
- The paper is well-written and mostly easy to follow

WEAKNESSES:
- The chosen setting is quite limited, since the authors focused specifically on 2-layers NNs and on modular addition, subtraction, and multiplication tasks. Therefore, it is not easy to understand if the given findings are generalizable
- The context of label noise is not the only context where memorization arises. Therefore, the authors should consider to extend their investigation and discussion to other memorization point of views, such as the memorization of atypical samples or rare sub-patterns of data
- Some of the choices behind the design of the experimental evaluation or the paper writing could be better justified. For example, the choice of the optimizers used and the design of the research questions which are left unanswered.

---

> ### Author Rebuttal · Authors · 2026-03-31
>
> We thank the reviewer for the positive assessment of our work, particularly regarding the importance of studying memorization-generalization entanglement and the clarity of our presentation. We appreciate the recognition of our extensive experimental evaluation and the relevance of our findings to the machine learning community. Below we address the specific questions raised.
>
> ## 1. Generalizability to larger models and more complex tasks
>
> We agree with the concerns regarding generalizability. We have conducted additional experiments to validate our results in more complex settings.
>
> ## 1.1 Result 3.2: noisy labels are memorized faster than clean labels
>
> > ... wonder if this ... due to the specific experimental settings ... . ... provide a more in-depth discussion ...
>
> This phenomenon generalizes to the MNIST dataset on a three-layer NN. With full-batch training, noisy accuracy at epoch 1 is significantly higher than clean accuracy:
>
> |         | **alpha=0.05, noisy** | **alpha=0.05, clean** | **alpha=0.2, noisy** | **alpha=0.2, clean** |
> | --------- | ----------------------------- | ----------------------------- | ---------------------------- | ---------------------------- |
> | epoch 1 | 10.87%                      | 2.86%                       | 10.75%                     | 2.89%                      |
> | epoch 2 | 6.47%                       | 46.29%                      | 5.71%                      | 48.09%                     |
>
> A key condition for this is full-batch training. If not, the gradient direction that benefits current batch of noisy labels is not transferable to another batch of noisy labels. We give a brief explanation fot this in the rebuttal to reviewer hjyW (due to the limited space here). We will add a detailed numerical gradient explanation and further discussion in the revised manuscript.
>
> ## 1.2 Frequency filtration improves generalization
>
> > ... frequency filtration ... be possible to filter out or reduce the effect of memorization in complex neural networks ...
>
> We applied frequency filtration to a Transformer by keeping only the top 20 frequencies in the unembedding head. Generalization improved significantly as the model forgot a large portion of label noise:
>
> | | clean acc | noisy acc | test acc|
> |---|---|---|---|
> | Before frequency filtration | 100% | 100% | 76.8% |
> | After frequency filtration | 98.1% | 36.8% | 87.8% |
>
> ## 1.3 Generalization gain by pruning
>
> > Is algorithm 1 and generally speaking the results of section 5 applicable to a larger NN model? ...
>
> For MNIST on MLP, we found that applying the pruning method from Section 5 caused both test and noisy accuracy to drop as the pruning ratio increased. This also supports our conclusion that standard training does not naturally separate noise from rules in structural level.
>
> ## 1.4 Other types of memorization and optimizers
>
> Our study focuses on explicit label noise to provide a clear boundary between structured generalization and unstructured memorization. This controlled setting is necessary for a precise mechanistic analysis. Investigating atypical samples or rare patterns is a compelling next step.
>
> Regarding SAM, it is specifically designed to bias models toward flatter minima. Our current work establishes a baseline mechanism under standard training. Characterizing how inductive biases like SAM alter these internal circuits deserves a dedicated study. we will list these in the future work part.
>
> ## 2. Justification of design and writing suggestions
>
> We will add the following justifications to the revised version:
>
> * Research Questions: Q1, Q2, and Q3 form the core logic of the paper and are addressed in Sections 3, 4, and 5, respectively. We will add explicit references to these questions in the corresponding sections.
> * Muon Optimizer: Muon was included to show that our results are consistent across different optimizers. Its behavior is similar to Adam and AdamW.
> * Figure 4 Clarity: The line types indicate the dataset. We will update the y-axis labels to 'train acc' and 'test acc' for better clarity.
>
> ## 3. Explanation of results
>
> ## 3.1 Usefulness of Result 4.1
>
> While analytical solutions exist for quadratic networks, they are missing for common ReLU networks. Our finding shows that ReLU networks generalize by imitating the frequency representations of quadratic models. This creates a theoretical bridge, allowing us to use analytical tools to study the behavior of ReLU models.
>
> ## 3.2 Multiplication task
>
> The reviewer is correct that Result 4.1 focuses on addition/subtraction. Modular multiplication over a prime P is isomorphic to modular addition over the group Z_{P-1} via discrete logarithms. The network can form its representation by learning the mapping in the log-domain, where the Fourier basis is mapped accordingly.

---

> > ### Author Rebuttal · Reviewer_ahKx · 2026-04-03
> >
> > I would like to thank the authors for their detailed rebuttal and the additional experiments provided. I am generally satisfied with the response, and my overall opinion of the paper remains more positive than negative. However, I am still somewhat on the fence. While I appreciate the effort to extend the experiments to MNIST and basic transformer components, these settings still fall short of the truly complex, large-scale computer vision or text processing tasks I originally inquired about, leaving the practical applicability of the proposed methods somewhat unproven for modern architectures. Furthermore, deferring the investigation of atypical samples and alternative optimizers like SAM to future work narrows the boundaries of your claims. Given these constraints, I believe the presentation of the results still requires careful refinement in the final manuscript to explicitly acknowledge these limitations and ensure the mechanistic insights are not overstated to the reader. Despite these reservations about the findings, the foundational analysis remains a valuable contribution to the community, and I will be maintaining my original score of a weak accept.

---

> > > ### Author Response · Authors · 2026-04-06
> > >
> > > We sincerely thank the reviewer for the positive evaluation and for recognizing the value of our foundational analysis. The feedback has been instrumental in refining our work.
> > >
> > > We fully agree regarding the task complexity. In the final manuscript, we will explicitly acknowledge these limitations and ensure the mechanistic insights are not overstated, clarifying the boundaries between our controlled study and large-scale applications.
> > >
> > > Regarding the SAM optimizer, during the discussion phase, we conducted additional experiments with radii $\rho \in \\{0.05, 0.1, 0.2\\}$ and noise ratios of $\\{0.1, 0.2, 0.3, 0.4\\}$. The results show that the model still perfectly memorizes label noise, and test accuracy remains non-decreasing as model width increases. This is consistent with our observations in Section 3, further confirming the robustness of the reported phenomena.
> > >
> > > Thanks again for the professional and constructive guidance.

---

### Decision · Program_Chairs · 2026-04-30

**Decision:**

Reject

**Comment:**

This paper studies the relation between memorization and generalization with modular arithmetic tasks and two-layer neural networks. It tackle an import research direction. However, most of the reviewers are concerned with the overly simplified setting of synthetic modular arithmetic task and more importantly simple 2-layer neural network experiments. The author added some additional but still simple settings such as MNIST experiments. The authors argued that the choice of 2-layer neural networks are because it is easy to study the memorization / generalization behaviors in this settings. However, the reviewers are not fully convinced that observations made in this case could robustly generalize to more general real world cases. Therefore, it is recommended to either add some in-depth theoretical analysis to justify the simplified settings, or to focus on more real world settings for empirical studies.